# EM-DARTS: Preventing Performance Collapse in Differentiable Architecture Search with The Edge Mutation Mechanism

## Abstract

Differentiable Architecture Search (DARTS) relaxes the discrete search space into a continuous form, significantly improving architecture search efficiency through gradient-based optimization. However, DARTS often suffers from performance collapse, where the performance of discovered architectures degrades during the search process, and the final architectures tend to be dominated by excessive skip-connections. In this work, we analyze how continuous relaxation impacts architecture optimization, identifying two main causes for performance collapse. First, the continuous relaxation framework introduces coupling between parametric operation weights and architecture parameters. This coupling leads to insufficient training of parametric operations, resulting in smaller architecture parameters for these operations. Second, DARTS's unrolled estimation property leads to larger architecture parameters for skip-connections. To attack this issue, we propose Edge Mutation Differentiable Architecture Search (EM-DARTS), where during network weight updates, edges have a probability of mutating from a weighted sum of candidate operations to a specific parametric operation. EM-DARTS reduces the impact of architecture parameters on parametric operations, allowing for better training of the parametric operations, thereby increasing their architecture parameters and preventing performance collapse. Theoretical results and experimental studies across diverse search spaces and datasets validate the effectiveness of the proposed method.

## 1 Introduction

Neural Architecture Search (NAS) has attracted considerable attention for its potential to automate and optimize the design of neural networks, which traditionally requires human expertise and extensive experimentation. Early NAS approaches were dominated by reinforcement learning and evolutionary algorithms (Zoph & Le, 2017; Real et al., 2017), which, though effective, were computationally expensive. In response, researchers proposed more efficient approaches, such as performance estimation techniques (Klein et al., 2016), network morphisms (Cai et al., 2018), and one-shot architecture search methods (Pham et al., 2018; Liu et al., 2019). Among these, one-shot methods stand out by leveraging weight sharing, enabling the training of a supernet encompassing all candidate sub-networks in a single pass.

Differentiable Architecture Search (DARTS) (Liu et al., 2019), a leading one-shot method, enhances efficiency by relaxing the discrete search space into a continuous one through architecture parameters. This enables gradient-based optimization of both the network weights and the architecture parameters in an alternating manner, making DARTS one of the most computationally efficient NAS approaches. Despite its advantages, DARTS is prone to performance collapse during the search process, as pointed by several studies (Zela et al., 2020; Chu et al., 2021). Specifically, the selected architectures are often dominated by excessive skip-connections, which reduce the representational capacity of the architectures and degrade performance. This issue is usually attributed to overfitting during the search process (Liang et al., 2020), the undue advantage of skip-connection (Chu et al., 2020; Xue et al., 2021; Ye et al., 2022), and limitations in weight-sharing frameworks (Movahedi et al., 2023), among other factors (Chen et al., 2019; Chen & Hsieh, 2020; Gu et al., 2021; Zhang et al., 2021).

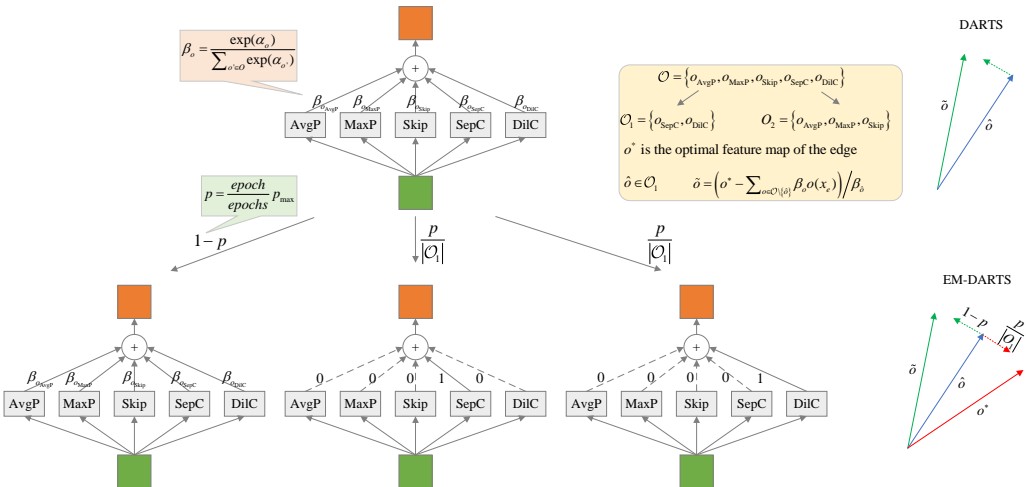

Figure 1: The edge mutation mechanism of EM-DARTS, where edges have a probability of mutating from a weighted sum of candidate operations to a specific parametric operation during network weight updates. Where $\alpha_o$ represents the architecture parameter of operation $o$, and $p_{\max}$ indicates the maximum probability of edges.

However, these DARTS alternatives overlook the impairment caused by the continuous relaxation framework on parametric operations.[1] Specifically, assuming that there is an optimal feature map for the output of an edge in the supernet, we found that the closer the output of the edge operation is to the optimal feature map, the larger the corresponding architecture parameter. However, the continuous relaxation framework introduces coupling between parametric operation weights and architecture parameters. By theoretical analysis, we show that this coupling causes the update targets of parametric operations to deviate from the direction of the optimal feature map, leading to insufficient training of these operations, where the distance between the operation output and the optimal feature map can hardly be minimized, resulting in smaller architecture parameters. Additionally, due to DARTS's unrolled estimation property (Greff et al., 2017; Wang et al., 2021), the output of skip-connection is closer to the optimal feature map, which results in larger architecture parameters for skip-connection. Consequently, the architecture parameters of parametric operations are frequently smaller than those of skip-connection, leading to the final searched architectures being dominated by an excessive skip-connections and resulting in poor performance.

Inspired by the sparse strategy of DSNAS (Hu et al., 2020), which can reduce the coupling between network weights and architectural parameters, we address this issue by introducing a mutation mechanism for edges in the DARTS supernet during network weight updates (see Figure 1). We term this method **E**dge **M**utation **D**ifferentiable **AR**chi**T**ecture **S**earch (EM-DARTS). EM-DARTS introduces randomness during the network weight update process, providing parametric operations with the opportunity to correct their update targets to the optimal feature map, effectively breaking the coupling between parametric operation weights and architecture parameters. Specifically, the mutation mechanism allows parametric operations to independently receive input data and perform forward and backward propagation. This process enables parametric operations to learn useful feature representations more effectively, optimize their architecture parameters, and prevent performance collapse. Additionally, as the mutation mechanism introduces negligible computational overhead, EM-DARTS preserves the efficiency of the DARTS search process. Our contributions are summarized as follows:

- We discover that that the performance collapse is primarily due to insufficient training of parameter operations, which results from the coupling issues between parametric operation weights and architecture parameters caused by the continuous relaxation framework.

---

[1]Candidate operations can be categorized into parametric and non-parametric operations based on whether they contain parameters. For example, parametric operation can be dilated convolution or separable convolution; non-parametric operation can be skip-connection, pooling, or no operation.

- We propose EM-DARTS and theoretically demonstrate that it allows for more thorough training of parametric operations, thereby preventing performance collapse.
- Extensive experiments on NAS-Bench-201, DARTS, and the reduced DARTS search spaces show that EM-DARTS achieves state-of-the-art performance, proving its effectiveness in addressing the performance collapse issue.

## 2 RELATED WORK

Several studies (Zela et al., 2020; Chu et al., 2021) have found that the performance of the architectures discovered during the DARTS search process tends to degrade continuously. To address this issue, researchers have proposed several improved DARTS methods. P-DARTS (Chen et al., 2019) addresses suboptimal architectures caused by the depth disparity between search and evaluation networks by gradually increasing the search network's depth. SmoothDARTS (Chen & Hsieh, 2020) identifies that a steep validation loss surface leads to sharp performance drops and smooths it through Hessian regularization, thereby improving search stability. DARTS+ (Liang et al., 2020) identifies the presence of overfitting during the architecture search process, which leads to performance degradation, and addresses this issue by introducing early stopping mechanisms. DOTS (Gu et al., 2021) observes that architecture parameters often fail to reflect the true importance of operations and introduces new evaluation and optimization strategies to improve search results. Likewise, IDARTS (Zhang et al., 2021) contends that the coupling of architecture parameters obscures their true importance, addressing this challenge through a backtracking method to manage different parameter types. Fair DARTS (Chu et al., 2020) identifies the unfair advantage of skip-connection in operation competition and mitigates it by using an independent sigmoid function to balance the weights of competing operations. DARTS- (Xue et al., 2021) reduces the influence of skip-connection by embedding auxiliary skip-connection within the cell design. Similarly, $\beta$-DARTS (Ye et al., 2022) introduces Beta-Decay regularization to limit the dominance of skip-connection, replacing the traditional $l_2$ regularization to improve balance. Recently, $\Lambda$-DARTS (Movahedi et al., 2023) finds that the weight-sharing framework limits DARTS' convergence by saturating the softmax function and improves convergence by aligning gradient layers to better harmonize operation selection, further stabilizing the search process. While these improvements offer valuable insights and partially mitigate performance collapse, they largely overlook how the continuous relaxation framework hampers parametric operations.

Meanwhile, SPOS (Guo et al., 2020) reduces computational resource usage by randomly selecting a single path for training within a supernet, maintaining the diversity of the search space and enhancing search efficiency. Building on this, GDAS (Dong & Yang, 2019) uses the Gumbel-Softmax distribution to make the search space continuous, allowing for gradient-based optimization of architecture parameters, thus further improving efficiency and stability. Finally, DSNAS (Hu et al., 2020) adds sparsity constraints to SPOS, directly optimizing network architectures and eliminating the need for parameter retraining, simplifying the search process. Inspired by the sparse strategy of DSNAS, which can reduce the coupling between network weights and architectural parameters, we propose EM-DARTS. However, unlike the aforementioned methods that primarily focus on improving search efficiency, EM-DARTS is specifically designed to address the issue of performance collapse in DARTS. By introducing an edge mutation mechanism, EM-DARTS aims to mitigate the adverse effects of the continuous relaxation framework on the training of parametric operations. This edge mutation mechanism can be seen as an extension of sparsity constraints. Because the purpose of using edge mutation is to enhance the training level of parametric operations, it differs from the sparsity strategies in SPOS and DSNAS. In EM-DARTS, edge mutations are restricted to parametric operations, rather than all operations. Moreover, EM-DARTS allows each edge to mutate into a specific parametric operation with a certain probability $p$, rather than enforcing each edge to mutate into a single operation.

## 3 METHOD

### 3.1 DIFFERENTIABLE ARCHITECTURE SEARCH OVERVIEW

The DARTS supernet consists of stacked normal and reduction cells, with each cell type sharing a set of architecture parameters. Each cell is structured as a directed acyclic graph (DAG) comprising

$N$ nodes $\{x_i\}_{i=0}^{N-1}$, where each node represents a feature map. An edge $(i, j)$ connects node $x_i$ to $x_j$, using $x_i$ as input. DARTS achieves continuous relaxation of the search space by introducing architecture parameters that represent each edge as a weighted combination of candidate operations from a set $\mathcal{O}$. The output of edge $(i, j)$ is defined as

$$\bar{o}^{(i,j)}(x_i) = \sum_{o \in \mathcal{O}} \beta_o^{(i,j)} o(x_i), \tag{1}$$

where $\beta_o^{(i,j)}$ is obtained by applying the softmax function to the vector $\alpha^{(i,j)} = \{\alpha_o^{(i,j)}\}$, i.e.,

$$\beta_o^{(i,j)} = \frac{\exp(\alpha_o^{(i,j)})}{\sum_{o' \in \mathcal{O}} \exp(\alpha_{o'}^{(i,j)})}, \tag{2}$$

and $\alpha_o^{(i,j)}$ is the architecture parameter for operation $o(\cdot)$ on edge $(i, j)$. The architecture parameters for all edges in a cell form its architecture parameter set $\alpha = \{\alpha^{(i,j)}\}$.

Each cell contains two input nodes, $N - 3$ intermediate nodes, and one output node. The input nodes are derived from the output nodes of the two preceding layer cells. Each intermediate node $x_j$ is computed as a sum of the outputs from all its predecessor nodes $x_i$ $(i < j)$, i.e.,

$$x_j = \sum_{i<j} \bar{o}^{(i,j)}(x_i). \tag{3}$$

The output node of the cell is obtained by concatenating the outputs of all intermediate nodes. To determine the optimal architecture parameters, DARTS alternates between gradient-based optimization of network weights $\omega$ and architecture parameters $\alpha$ by solving the following bi-level optimization problem, i.e.,

$$\begin{aligned} \min_{\alpha} \quad & \mathcal{L}_{val}(\omega^*(\alpha), \alpha) \\ \text{s.t.} \quad & \omega^*(\alpha) = \arg\min_{\omega} \mathcal{L}_{train}(\omega, \alpha), \end{aligned} \tag{4}$$

where $\mathcal{L}_{train}$ and $\mathcal{L}_{val}$ denote the training and validation loss functions, respectively. DARTS proposes two methods to approximate $\omega^*(\alpha)$: the first-order and second-order methods. In our work, we utilize the first-order method. For further details on DARTS, please refer to the original paper (Liu et al., 2019).

## 3.2 THEORETICAL ANALYSIS OF PERFORMANCE COLLAPSE

DARTS introduces architecture parameters to continuously relax the search space, allowing us to select the optimal operations based on the magnitude of these parameters. However, the introduced architecture parameters form a coupling between parametric operation weights and architecture parameters, which affects the training of parametric operations and makes the architecture optimization process more complex. In this section, we will provide a theoretical analysis of the impact of continuous relaxation on architecture optimization.

As indicated by DARTS-PT (Wang et al., 2021), DARTS exhibits an unrolled estimation property, where all edges within a cell attempt to estimate the same optimal feature map. For simplicity, we assume that when DARTS converges, the distance between edges' outputs within each cell and the cell's optimal feature map is minimized. Since the outputs of operations on each edge in DARTS are normalized to the same scale, both the edge output and the optimal feature map of the cell should also have the same scale. Therefore, the distance between the operation output and the optimal feature map can be represented by variance of the difference between them, and the distance between the edge output and the optimal feature map can be similarly applied.[2] In the following proposition, we focus on an edge of a supernet cell and analyze the properties of the architecture parameters on this edge under the continuous relaxation framework, when DARTS converges.

**Proposition 1.** *Let $(i, j)$ be an edge in a supernet cell and $o^*$ be the optimal feature map on this cell. Under the assumptions that (1) weight-sharing has no impact on the architecture parameters, and (2) the DARTS algorithm is convergent, then the architecture parameter $\alpha_o^{(i,j)}$ is approximately*

---

[2]When variables $X$ and $Y$ have the same scale, then $E[X] = E[Y]$, and the variance of the difference between them is $\text{Var}(X - Y) = E[(X - Y)^2]$.

*inversely proportional to the variance of the difference between the operation output $o(x_i)$ and the optimal feature map $o^*$. That is,*

$$\alpha_o^{(i,j)} \propto \frac{1}{Var(o(x_i) - o^*)}. \tag{5}$$

Its proof is postponed to Appendix A.1. According to Proposition 1, the smaller the distance $Var(o(x_i) - o^*)$ between the operation output and the optimal feature map, the larger the corresponding architecture parameter $\alpha_o^{(i,j)}$ will be. However, parametric operations can reduce the distance between their output and the optimal feature map by adjusting their weights, thereby improving their architecture parameters. In contrast, non-parametric operations lack this flexibility, and their distance remains fixed. Therefore, to ensure that the architecture parameters accurately reflect the importance of each operation, parametric operations must be fully trained to minimize the distance between their output and the optimal feature map. In the following proposition, we will evaluate the training of parametric operations under the condition of DARTS convergence.

**Proposition 2.** *Let $(i, j)$ be an edge in a supernet cell and $o^*$ be the optimal feature map on this cell. Define $\mathcal{O}_1$ as the set of all parametric operations in $\mathcal{O}$. Assuming DARTS converges, then, for $o \in \mathcal{O}_1$, the variance of the difference between the parametric operation's output $o(x_i)$ and the optimal feature map $o^*$ does not reach its minimum value.*

See Appendix A.2 for a detailed proof. Proposition 2 indicates that due to the coupling introduced by continuous relaxation, parametric operations cannot be fully trained, which is a critical flaw of the continuous relaxation framework. In fact, for any $o \in \mathcal{O}_1$, the goal of the parametric operation $o$ is not to be fully trained itself, but to contribute to the overall performance of the edge. That is, $o(x_i)$ converges not to the optimal feature map $o^*$, but to a shifted target $\tilde{o}$ influenced by other operations, i.e.,

$$\tilde{o} = \frac{o^* - \sum_{o' \in \mathcal{O} \setminus \{o\}} \beta_{o'}^{(i,j)} o'(x_i)}{\beta_o^{(i,j)}}, \tag{6}$$

where $\beta_o^{(i,j)}$ is defined in Equation (2). In addition, since the output of the skip-connection $x_i$ is derived from the mixed output of the previous edges, and each edge estimates $o^*$, $x_i$ directly approximates $o^*$. Consequently, this results in a very small $Var(x_i - o^*)$, making the architecture parameters of the skip-connection larger. As a result, the architecture parameters of parametric operations are frequently smaller than those of the skip-connection, leading to the final searched architectures being dominated by an excessive number of skip-connections and resulting in performance collapse.

### 3.3 Edge Mutation Differentiable Architecture Search

To preserve DARTS' efficiency and address performance collapse, we propose a novel method of introducing a mutation mechanism on the edges during network weight updates, and at the same time preserving the edge states during architectural parameter updates. This mechanism assigns a probability to edges, allowing the output of a weighted sum of candidate operations to mutate into that of a specific parametric operation. Therefore, during network weight updates, the output of edge $(i, j)$ is calculated using the following formula,

$$\bar{o}^{(i,j)} = \sum_{o \in \mathcal{O}} \left( \gamma^{(i,j)} \hat{\beta}_o^{(i,j)} + (1 - \gamma^{(i,j)}) \beta_o^{(i,j)} \right) o(x_i), \tag{7}$$

where $\gamma^{(i,j)}$ and $\hat{\beta}_o^{(i,j)}$ are the mutation factor and mutation weight, respectively.[3] The mutation factor $\gamma^{(i,j)}$ follows a Bernoulli distribution with probability $p$, with its probability mass function is

$$P(\gamma^{(i,j)} = k) = \begin{cases} p, & \text{if} \quad k = 1, \\ 1 - p, & \text{if} \quad k = 0. \end{cases} \tag{8}$$

The mutation weight $\hat{\beta}_o^{(i,j)}$ is given by

$$\begin{cases} \hat{\beta}_{\hat{o}}^{(i,j)} = 1, & \hat{o} \xleftarrow{\text{rand}} \mathcal{O}_1, \\ \hat{\beta}_{o'}^{(i,j)} = 0, & o' \in \mathcal{O} \setminus \{\hat{o}\}, \end{cases} \tag{9}$$

---

[3]The values of $\gamma^{(i,j)}$ and $\hat{\beta}_o^{(i,j)}$ are unique across different layers.

where $\hat{o} \xleftarrow{\text{rand}} \mathcal{O}_1$ means that $\hat{o}$ is randomly selected from the set of parametric operations $\mathcal{O}_1$. So, the bi-level optimization process of architecture parameters $\alpha$ and network weights $\omega$ is modified in the following way,

$$
\begin{aligned}
\min_{\alpha} \quad & \mathcal{L}_{val}(\omega^*(\alpha), \alpha) \\
\text{s.t.} \quad & \omega^*(\alpha) = \arg\min_{\omega} \mathcal{L}_{train}(\omega, \gamma, \hat{\beta}),
\end{aligned}
\tag{10}
$$

where $\gamma$ represents the collection of mutation factors for all edges, and $\hat{\beta}$ represents the mutation weights for all operations. For the proposed method, increasing the edge mutation probability $p$ provides parametric operations with more opportunities to align with the optimal feature map $o^*$. However, if $p$ is chosen to be too large, it will lead to an increase in the number of edges that mutate to have only one parametric operation, thereby causing drastic structural changes in the supernet and potentially destabilizing network weight updates. Conversely, if $p$ is set too small, the training improvement for parametric operations will be limited, still posing a risk of performance collapse. Previous studies (Zela et al., 2020; Chu et al., 2021) have shown that in DARTS, the architecture performance typically begins to decline in the middle of the search process, and this degradation becomes more severe as training progresses. This indicates that although parametric operations can initially approach the optimal feature map $o^*$ to some extent, they gradually deviate from $o^*$ as the search progresses due to the shifting of their update targets. Therefore, we suggest gradually increasing support for parametric operations as the search progresses. To achieve this, we propose a growth strategy where the mutation probability $p$ starts at 0 and increases linearly during the search process until it reaches its maximum value $p_{\max}$. The details of this approach are summarized in Algorithm 1.

---

**Algorithm 1:** EDGE MUTATION DIFFERENTIABLE ARCHITECTURE SEARCH

**Input:** Training data, validation data, search hyper-graph, hyper-parameters $p_{\max}$
**Output:** Network architecture
Create architecture parameters $\alpha$; set $t = 1$, $T = 50$
**while** $t \leq T$ **do**
    1. Update architecture parameters $\alpha$ by descending $\nabla_{\alpha}\mathcal{L}_{val}(\omega, \alpha)$
    2. Set $p \leftarrow \frac{t}{T}p_{\max}$
    3. Compute $\gamma$ and $\hat{\beta}$ using Equations (8) and (9), respectively
    4. Update network weights $\omega$ by descending $\nabla_{\omega}\mathcal{L}_{train}(\omega, \gamma, \hat{\beta})$
    5. Set $t \leftarrow t + 1$

Derive the final architecture based on the optimized $\alpha$

---

The edge mutation mechanism effectively breaks the coupling between parametric operation weights and architecture parameters by introducing randomness during the network weight update process of the supernet. Specifically, during each network weight update, each edge has a certain probability $p$ of mutating into a specific parametric operation, rather than continuing to rely on the weighted average of all candidate operations. This mutation process grants parametric operations the opportunity to independently receive input data and perform forward and backward propagation, facilitating more effective learning of useful feature representations. Through this method, parametric operations can better approximate optimal feature maps, thereby optimizing their architecture parameters. The following theorem validates the effectiveness of the edge mutation mechanism in improving the overall training of parametric operations.

**Theorem 1.** *Let $(i, j)$ be an edge in a supernet cell and $o^*$ be the optimal feature map on this cell. Define $\mathcal{O}_1$ as the set of all parametric operations in $\mathcal{O}$, with $o_1, o_2, \ldots, o_s \in \mathcal{O}_1$ having parameters $\omega_1, \ldots, \omega_s$. Assuming DARTS converges, the parameters of these operations and the architecture parameters for edge $(i, j)$ are $\widehat{\omega}_1, \widehat{\omega}_2, \ldots, \widehat{\omega}_s$ and $\widehat{\alpha}$, respectively. Additionally, if EM-DARTS converges, its parameters are $\tilde{\omega}_1, \tilde{\omega}_2, \ldots, \tilde{\omega}_s$ and $\tilde{\alpha}$. Then we have the following inequality,*

$$
\sum_{i=1}^{s} Var(o_i(x_i, \tilde{\omega}_i) - o^*) < \sum_{i=1}^{s} Var(o_i(x_i, \widehat{\omega}_i) - o^*).
\tag{11}
$$

The proof is provided in Appendix A.3. Theorem 1 demonstrates that EM-DARTS achieves better training of parametric operations compared to DARTS. Actually, EM-DARTS provides parametric

operations with an opportunity to correct their update targets to the optimal feature map $o^*$, as shown in Figure 1. This reduces the interference from architectural parameters on the parametric operations, allowing them to be trained more thoroughly, which in turn reduces the distance between their output and $o^*$. Consequently, this increases the architecture parameters of the parametric operations, making them larger than those of the skip-connection, thus avoiding performance collapse.

# 4 EXPERIMENTAL STUDIES

In this section, we evaluate the effectiveness of EM-DARTS through a series of experiments on the NAS-Bench-201 and DARTS search space (Dong & Yang, 2020; Liu et al., 2019). We test its robustness in the reduced DARTS search space (Zela et al., 2020) across various datasets. In addition, we conduct an ablation study within the NAS-Bench-201 search space to further validate the effectiveness of EM-DARTS, demonstrate the robustness of growth strategy, and explore the impact of various hyperparameters. The details of the datasets, search costs, experimental settings, and discovered architectures are provided in Appendices A.4, A.6, A.5, and A.8, respectively. In addition, we define a search space $S5$ that contains only parametric operations, and validate the effectiveness of EM-DARTS in this search space. See Appendix A.7 for details.

## 4.1 NAS-BENCH-201 SEARCH SPACE

We began by evaluating the effectiveness of EM-DARTS in the NAS-Bench-201 search space. As shown in Table 1, EM-DARTS demonstrates significant performance improvements compared to other DARTS-based algorithms. It consistently achieves performance on par with state-of-the-art methods such as $\Lambda$-DARTS and $\beta$-DARTS (Movahedi et al., 2023; Ye et al., 2022).

Table 1: Performance comparison on the NAS-Bench-201 benchmark. EM-DARTS are conducted by searching on the CIFAR-10 dataset and evaluating on CIFAR-10, CIFAR-100, and ImageNet16-120. The reported accuracy values are the mean and standard deviation derived from 4 independent runs. (1st): first-order; (2nd): second-order.

| Method | CIFAR-10 | | CIFAR-100 | | ImageNet16-120 | |
|---|---|---|---|---|---|---|
| | Valid | Test | Valid | Test | Valid | Test |
| DARTS(1st) (Liu et al., 2019) | $39.77 \pm 0.00$ | $54.30 \pm 0.00$ | $15.03 \pm 0.00$ | $15.61 \pm 0.00$ | $16.43 \pm 0.00$ | $16.32 \pm 0.00$ |
| DARTS(2nd) (Liu et al., 2019) | $39.77 \pm 0.00$ | $54.30 \pm 0.00$ | $15.03 \pm 0.00$ | $15.61 \pm 0.00$ | $16.43 \pm 0.00$ | $16.32 \pm 0.00$ |
| GDAS (Dong & Yang, 2019) | $89.89 \pm 0.08$ | $93.61 \pm 0.09$ | $71.34 \pm 0.04$ | $70.70 \pm 0.30$ | $41.59 \pm 1.33$ | $41.71 \pm 0.98$ |
| SNAS (Xie et al., 2019) | $90.10 \pm 1.04$ | $92.77 \pm 0.83$ | $69.69 \pm 2.39$ | $69.34 \pm 1.98$ | $42.84 \pm 1.79$ | $43.16 \pm 2.64$ |
| DSNAS (Hu et al., 2020) | $89.66 \pm 0.29$ | $93.08 \pm 0.13$ | $30.87 \pm 16.40$ | $31.01 \pm 16.38$ | $40.61 \pm 0.09$ | $41.07 \pm 0.09$ |
| PC-DARTS (Xu et al., 2020) | $89.96 \pm 0.15$ | $93.41 \pm 0.30$ | $67.12 \pm 0.39$ | $67.48 \pm 0.89$ | $40.83 \pm 0.08$ | $41.31 \pm 0.22$ |
| iDARTS (Zhang et al., 2021) | $89.86 \pm 0.60$ | $93.58 \pm 0.32$ | $70.57 \pm 0.24$ | $70.83 \pm 0.48$ | $40.38 \pm 0.59$ | $40.89 \pm 0.68$ |
| DARTS- (Chu et al., 2021) | $91.03 \pm 0.44$ | $93.80 \pm 0.40$ | $71.36 \pm 1.51$ | $71.53 \pm 1.51$ | $44.87 \pm 1.46$ | $45.12 \pm 0.82$ |
| VIM-NAS (Yaoming et al., 2021) | $91.48 \pm 0.09$ | $94.31 \pm 0.11$ | $73.12 \pm 0.51$ | $73.07 \pm 0.58$ | $45.92 \pm 0.51$ | $46.27 \pm 0.17$ |
| DrNAS (Chen et al., 2021) | $\mathbf{91.55 \pm 0.00}$ | $\mathbf{94.36 \pm 0.00}$ | $\mathbf{73.49 \pm 0.00}$ | $\mathbf{73.51 \pm 0.00}$ | $\mathbf{46.37 \pm 0.00}$ | $\mathbf{46.34 \pm 0.00}$ |
| $\beta$-DARTS (Ye et al., 2022) | $\mathbf{91.55 \pm 0.00}$ | $\mathbf{94.36 \pm 0.00}$ | $\mathbf{73.49 \pm 0.00}$ | $\mathbf{73.51 \pm 0.00}$ | $\mathbf{46.37 \pm 0.00}$ | $\mathbf{46.34 \pm 0.00}$ |
| $\Lambda$-DARTS (Movahedi et al., 2023) | $\mathbf{91.55 \pm 0.00}$ | $\mathbf{94.36 \pm 0.00}$ | $\mathbf{73.49 \pm 0.00}$ | $\mathbf{73.51 \pm 0.00}$ | $\mathbf{46.37 \pm 0.00}$ | $\mathbf{46.34 \pm 0.00}$ |
| **EM-DARTS** | $\mathbf{91.55 \pm 0.00}$ | $\mathbf{94.36 \pm 0.00}$ | $\mathbf{73.49 \pm 0.00}$ | $\mathbf{73.51 \pm 0.00}$ | $\mathbf{46.37 \pm 0.00}$ | $\mathbf{46.34 \pm 0.00}$ |
| Optimal (Dong & Yang, 2020) | 91.61 | 94.37 | 73.49 | 73.51 | 46.77 | 47.31 |

## 4.2 DARTS SEARCH SPACE

To verify its effectiveness in preventing performance collapse, we tested EM-DARTS in the original DARTS search space (Liu et al., 2019) across multiple datasets. Table 2 shows that EM-DARTS substantially outperforms baseline models. Specifically, on the CIFAR-10, CIFAR-100, and ImageNet datasets, EM-DARTS achieves average accuracies of 97.62%, 83.96%, and 76.2%, respectively, surpassing current state-of-the-art methods by 0.05%, 0.11%, and 0.2%.

## 4.3 REDUCED SEARCH SPACES

We conducted experiments in the Reduced DARTS search space, as proposed by R-DARTS (Zela et al., 2020), to validate the robustness of EM-DARTS. As shown in Table 3, EM-DARTS outperforms the current state-of-the-art methods in 9 out of 12 experiments, while performing comparably

Table 2: Performance comparison on the DARTS benchmark. The first block reports the best performance of the architecture, whereas the second block reports the average performance of multiple searches, except for results on ImageNet. EM-DARTS are conducted by searching on the CIFAR-10 dataset and evaluating on CIFAR-10, CIFAR-100, and ImageNet. The reported accuracy values are the mean and standard deviation derived from 4 independent runs.
[†]: Searching on ImageNet.

| Method | CIFAR-10 | | CIFAR-100 | | ImageNet | |
|---|---|---|---|---|---|---|
| | Params (M) | Test Acc (%) | Params (M) | Test Acc (%) | Params (M) | Test Acc (%) |
| NASNet-A (Zoph et al., 2018) | 3.3 | 97.35 | 3.3 | 83.18 | 5.3 | 74.0 |
| DARTS(1st) (Liu et al., 2019) | 3.4 | $97.00 \pm 0.14$ | 3.4 | 82.46 | - | - |
| DARTS(2nd) (Liu et al., 2019) | 3.3 | $97.24 \pm 0.09$ | - | - | 4.7 | 73.3 |
| SNAS (Xie et al., 2019) | 2.8 | $97.15 \pm 0.02$ | 2.8 | 82.45 | 4.3 | 72.7 |
| GDAS (Dong & Yang, 2019) | 3.4 | 97.07 | 3.4 | 81.62 | 5.3 | 74.0 |
| P-DARTS (Chen et al., 2019) | 3.4 | 97.50 | 3.6 | 82.51 | 5.1 | 75.3 |
| PC-DARTS (Xu et al., 2020) | 3.6 | $97.43 \pm 0.07$ | 3.6 | **83.10** | 5.3 | 75.8 |
| DrNAS (Chen et al., 2021) | 4.0 | $97.46 \pm 0.03$ | - | - | 5.2 | 75.8 |
| VIM-NAS (Yaoming et al., 2021) | 3.9 | $\mathbf{97.55 \pm 0.04}$ | - | - | - | **76.0** |
| SWAP-NAS (Peng et al., 2024) | 4.3 | $97.52 \pm 0.04$ | - | - | 5.8 | **76.0** |
| R-DARTS (Zela et al., 2020) | - | $97.05 \pm 0.21$ | - | $81.99 \pm 0.26$ | - | - |
| P-DARTS (Chen et al., 2019) | $3.3 \pm 0.21$ | $97.19 \pm 0.14$ | - | - | - | - |
| SDARTS-ADV (Chen & Hsieh, 2020) | 3.3 | $97.39 \pm 0.02$ | - | - | 5.4 | 74.8 |
| DOTS (Gu et al., 2021) | 3.5 | $97.51 \pm 0.06$ | 4.1 | $83.52 \pm 0.13$ | 5.2 | 75.7 |
| DARTS-PT (Wang et al., 2021) | 3.0 | $97.39 \pm 0.08$ | - | - | 4.6 | 74.5 |
| DARTS- (Chu et al., 2021) | $3.5 \pm 0.13$ | $97.41 \pm 0.08$ | 3.4 | $82.49 \pm 0.25$ | 4.9 | $76.2^{\dagger}$ |
| $\beta$-DARTS (Ye et al., 2022) | $3.8 \pm 0.08$ | $97.49 \pm 0.07$ | $3.8 \pm 0.08$ | $83.48 \pm 0.03$ | 5.4 | 75.8 |
| $\Lambda$-DARTS (Movahedi et al., 2023) | $3.6 \pm 0.13$ | $\mathbf{97.57 \pm 0.05}$ | $3.6 \pm 0.1$ | $\mathbf{83.85 \pm 0.38}$ | 3.8 | 75.7 |
| **EM-DARTS (avg)** | $4.3 \pm 0.1$ | $\mathbf{97.62 \pm 0.05}$ | $4.4 \pm 0.1$ | $\mathbf{83.96 \pm 0.19}$ | - | - |
| **EM-DARTS (best)** | 4.4 | **97.67** | 4.5 | **84.19** | 6.2 | **76.2** |

in the remaining two. These results confirm that EM-DARTS is highly effective at preventing performance collapse across different environments.

Table 3: Performance comparison on the Reduced DARTS benchmark. The reported test error rate (%) is from the best-performing architecture of 4 independent runs.

| Dataset | Search Space | DARTS | PC-DARTS | R-DARTS | SDARTS-RS | $\Lambda$-DARTS | EM-DARTS |
|---|---|---|---|---|---|---|---|
| CIFAR-10 | S1 | 3.84 | 3.11 | 2.78 | 2.78 | 2.83 | **2.74** |
| | S2 | 4.85 | 3.02 | 3.31 | 2.75 | 2.56 | **2.39** |
| | S3 | 3.34 | 2.51 | 2.51 | 2.53 | **2.38** | 2.44 |
| | S4 | 7.20 | 3.02 | 3.56 | 2.93 | 2.46 | **2.32** |
| CIFAR-100 | S1 | 29.46 | 24.25 | 23.51 | 24.48 | **22.79** | 23.37 |
| | S2 | 26.05 | 22.48 | 22.44 | 22.28 | 21.68 | **21.47** |
| | S3 | 28.90 | 21.69 | 23.99 | 21.09 | 21.03 | **20.33** |
| | S4 | 22.85 | 21.50 | 21.94 | 21.46 | 20.65 | **20.35** |
| SVHN | S1 | 4.58 | 2.47 | **2.35** | 2.62 | 2.39 | 2.49 |
| | S2 | 3.53 | 2.42 | 2.51 | 2.39 | 2.37 | **2.34** |
| | S3 | 3.41 | 2.41 | 2.48 | 2.36 | 2.31 | **2.30** |
| | S4 | 3.05 | 2.43 | 2.50 | 2.46 | 2.34 | **2.27** |

## 4.4 ABLATION STUDY

To further verify the effectiveness of the edge mutation mechanism and the probability growth strategy, we conducted several comparative experiments. We set up three phases of edge mutation: early phase with a mutation probability of 0.2 for the first 1/3 of epochs, middle phase with a mutation probability of 0.2 for the next 1/3 of epochs, and late phase with a mutation probability of 0.2 for the last 1/3 of epochs. We compared these settings with fixed mutation probabilities of 0.1 and 0.2, as well as a linearly increasing probability from 0 to 0.2. To assess the robustness of the probability growth strategy, we compared linear, exponential, and cosine growth strategies. We also performed experiments with different $p_{\max}$ values (0.1, 0.2, 0.3, 0.4, and 0.5). Finally, to further explore

Table 4: Performance variation of different growth strategies. The experiment is conducted in the NAS-Bench-201 benchmark using CIFAR-10. The mean and standard deviation of the accuracy are calculated over 4 independent runs.

| Strategy | Accuracy (%) |
|---|---|
| **linear** | $\mathbf{91.55 \pm 0.00}$ |
| exponential | $91.52 \pm 0.02$ |
| cosine | $91.53 \pm 0.03$ |

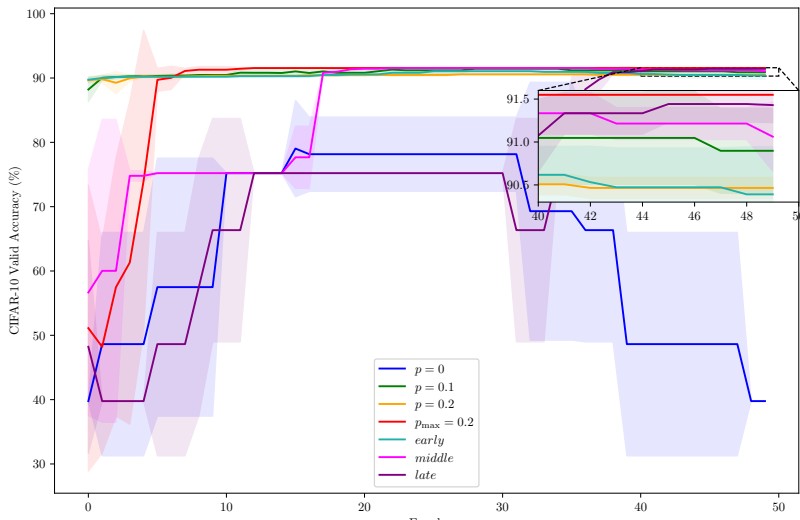

Figure 2: Performance variation of the architecture during the search process. The experiment is conducted in the NAS-Bench-201 benchmark using CIFAR-10. Results are averaged over 4 runs and presented with a 95% confidence interval.

the capability of EM-DARTS in preventing performance collapse, we extended the search process to 400 epochs and rigorously evaluated the discovered architectures' performance.

Figure 2 shows that when the mutation strategy is not applied, performance collapse occurs again. Interestingly, all methods that use the edge mutation mechanism successfully avoid performance collapse, with the final architectures achieving accuracies exceeding 90%, highlighting the effectiveness of the mutation mechanism. Additionally, we find that using a probability growth strategy ensures that the optimal architecture is found early and remains stable. These findings emphasize the necessity of the growth strategy. Table 4 shows that all growth strategies yield similar results, confirming the overall robustness of the approach. However, the linear growth strategy slightly outperforms the others and is simpler to implement, so we recommend using it.

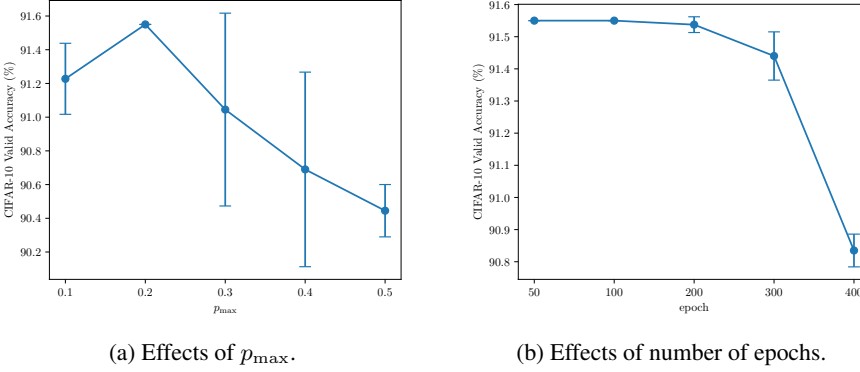

(a) Effects of $p_{max}$.      (b) Effects of number of epochs.

Figure 3: The impact of hyperparameters on EM-DARTS performance is analyzed. The experiment is conducted in the NAS-Bench-201 benchmark using CIFAR-10. Results are averaged over four runs and presented with a 95% confidence interval.

Figure 3(a) shows that as $p_{max}$ increases, performance initially improves, peaking at $p_{max} = 0.2$ before declining. This trend reflects that the hyperparameter $p_{max}$ needs to find a balance between reducing the interference of architecture parameters on parametric operations and avoiding instabil-

ity in the supernet structure. Figure 3(b) shows that extending training to 400 epochs results in only a slight 0.7% drop in performance, with optimal performance maintained at 100 epochs and a minor 0.1% decline by 300 epochs. This demonstrates that EM-DARTS effectively prevents performance degradation over extended training periods.

## 5 CONCLUSION

In response to the issue of performance collapse caused by the continuous relaxation framework, we propose the EM-DARTS method, which probabilistically mutates the edges of the DARTS supernet from weighted combinations of candidate operations to a specific parametric operation during network weight updates. This method effectively mitigates the issue of performance collapse, leading to more robust and efficient network architectures, which is supported by both theoretical analysis and extensive experimental validation. Although EM-DARTS successfully addresses the problem of performance degradation, the effectiveness of this method heavily depends on the setting of the mutation probability. An important direction for future research is to explore search frameworks that can break through the limitation of dependence on the setting.

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

# A    APPENDIX

## A.1    PROOF OF PROPOSITION 1

*Proof.* The variance of the difference between $\bar{o}^{(i,j)}(x_i)$ and $o^*$ can be expressed as

$$\text{Var}(\bar{o}^{(i,j)}(x_i) - o^*) = \text{Var}\left(\sum_{o \in \mathcal{O}} \beta_o^{(i,j)} o(x_i) - o^*\right). \tag{12}$$

Expanding the variance term, we have

$$\text{Var}(\bar{o}(x_i) - o^*) = \sum_{o \in \mathcal{O}} \beta_o^{(i,j)^2} \text{Var}(o(x_i) - o^*)$$
$$+ \sum_{o \in \mathcal{O}} \sum_{o' \in \mathcal{O} \setminus \{o\}} \beta_o^{(i,j)} \beta_{o'}^{(i,j)} \text{Cov}(o(x_i) - o^*, o'(x_i) - o^*). \tag{13}$$

Empirically, $\text{Cov}(o(x_i) - o^*, o'(x_i) - o^*)$ is relatively small compared to $\text{Var}(o(x_i) - o^*)$, and retaining it would considerably complicate subsequent calculations. Therefore, we assume that $\text{Cov}(o(x_i) - o^*, o'(x_i) - o^*) \approx 0$. Thus, we have

$$\text{Var}(\bar{o}^{(i,j)}(x_i) - o^*) \approx \sum_{o \in \mathcal{O}} \beta_o^{(i,j)^2} \text{Var}(o(x_i) - o^*). \tag{14}$$

When DARTS converges, $\text{Var}(\bar{o}^{(i,j)}(x_i) - o^*)$ reaches its minimum value. Additionally, the weights $\beta_o^{(i,j)}$ must satisfy the constraint $\sum_{o \in \mathcal{O}} \beta_o^{(i,j)} = 1$. To solve for $\beta_o^{(i,j)}$ at DARTS convergence, we use the Lagrangian method,

$$\mathcal{L}(\beta_{o_1}^{(i,j)}, \beta_{o_2}^{(i,j)}, \ldots, \beta_{o_{|\mathcal{O}|}}^{(i,j)}, \lambda) = \sum_{o \in \mathcal{O}} \beta_o^{(i,j)^2} \text{Var}(o(x_i) - o^*) - \lambda \left(\sum_{o \in \mathcal{O}} \beta_o^{(i,j)} - 1\right), \tag{15}$$

where $\lambda$ is the Lagrange multiplier enforcing the constraint. Taking the derivative of $\mathcal{L}$ with respect to $\beta_o^{(i,j)}$ and setting it to zero, we obtain

$$\frac{\partial \mathcal{L}}{\partial \beta_o^{(i,j)}} = 2\beta_o^{(i,j)} \text{Var}(o(x_i) - o^*) - \lambda = 0, \tag{16}$$

which simplifies to

$$\beta_o^{(i,j)} \text{Var}(o(x_i) - o^*) = \frac{\lambda}{2}. \tag{17}$$

Applying the constraint $\sum_{o \in \mathcal{O}} \beta_o^{(i,j)} = 1$ and solving for $\lambda$, we find

$$\lambda = 2 \left( \sum_{o \in \mathcal{O}} \frac{1}{\text{Var}(o(x_i) - o^*)} \right)^{-1}. \tag{18}$$

Substituting $\lambda$ back into the expression for $\beta_o^{(i,j)}$, we obtain

$$\beta_o^{(i,j)} = \frac{\frac{1}{\text{Var}(o(x_i) - o^*)}}{\sum_{o' \in \mathcal{O}} \frac{1}{\text{Var}(o'(x_i) - o^*)}}. \tag{19}$$

Since $\beta_o^{(i,j)}$ is defined as $\frac{\exp(\alpha_o^{(i,j)})}{\sum_{o' \in \mathcal{O}} \exp(\alpha_{o'}^{(i,j)})}$, it follows that

$$\exp(\alpha_o^{(i,j)}) \propto \frac{1}{\text{Var}(o(x_i) - o^*)}. \tag{20}$$

Therefore, we conclude that

$$\alpha_o^{(i,j)} \propto \frac{1}{\text{Var}(o(x_i) - o^*)}. \tag{21}$$

$\square$

## A.2 PROOF OF PROPOSITION 2

*Proof.* Consider that the variance $\text{Var}(\bar{o}(x_i) - o^*)$ can be expanded as

$$\text{Var}(\bar{o}^{(i,j)}(x_i) - o^*) = \sum_{o \in \mathcal{O}} \beta_o^{(i,j)2} \text{Var}(o(x_i) - o^*)$$
$$+ \sum_{o \in \mathcal{O}} \sum_{o' \in \mathcal{O} \setminus \{o\}} \beta_o^{(i,j)} \beta_{o'}^{(i,j)} \text{Cov}(o(x_i) - o^*, o'(x_i) - o^*). \tag{22}$$

For any $o \in \mathcal{O}_1$ with parameter $\omega_o$, the derivative of this variance with respect to $\omega_o$ is

$$\frac{\partial \text{Var}(\bar{o}^{(i,j)}(x_i) - o^*)}{\partial \omega_o} = \beta_o^{(i,j)2} \frac{\partial \text{Var}(o(x_i) - o^*)}{\partial \omega_o}$$
$$+ \beta_o^{(i,j)} \sum_{o' \in \mathcal{O} \setminus \{o\}} \beta_{o'}^{(i,j)} \frac{\partial \text{Cov}(o(x_i) - o^*, o'(x_i) - o^*)}{\partial \omega_o}. \tag{23}$$

Since DARTS, at convergence, minimizes $\text{Var}(\bar{o}^{(i,j)}(x_i) - o^*)$, we have

$$\frac{\partial \text{Var}(\bar{o}^{(i,j)}(x_i) - o^*)}{\partial \omega_o} = \mathbf{0}. \tag{24}$$

This leads to

$$\beta_o^{(i,j)} \frac{\partial \text{Var}(o(x_i) - o^*)}{\partial \omega_o} = - \sum_{o' \in \mathcal{O} \setminus \{o\}} \beta_{o'}^{(i,j)} \frac{\partial \text{Cov}(o(x_i) - o^*, o'(x_i) - o^*)}{\partial \omega_o}. \tag{25}$$

Since

$$\sum_{o' \in \mathcal{O} \setminus \{o\}} \beta_{o'}^{(i,j)} \frac{\partial \text{Cov}(o(x_i) - o^*, o'(x_i) - o^*)}{\partial \omega_o} \neq \mathbf{0}, \tag{26}$$

which implies that

$$\frac{\partial \text{Var}(o(x_i) - o^*)}{\partial \omega_o} \neq \mathbf{0}. \tag{27}$$

Therefore, the convergence of DARTS does not necessarily guarantee that the variance $\text{Var}(o(x_i) - o^*)$ for each $o \in \mathcal{O}_1$ is minimized. $\square$

### A.3 PROOF OF THEOREM 1

*Proof.* In EM-DARTS, the expected value of the objective function for parametric operations is given by

$$f(\omega_1, \ldots, \omega_s, \alpha) = \frac{p_{\max}}{s} \sum_{i=1}^{s} \text{Var}(o_i(x_i, \omega_i) - o^*)$$
$$+ (1 - p_{\max})\text{Var}(\bar{o}(x_i, \omega_1, \ldots, \omega_s, \alpha) - o^*), \tag{28}$$

Since $f$ reaches its minimum at $\tilde{\omega}_1, \ldots, \tilde{\omega}_s, \tilde{\alpha}$, we have the following inequality

$$\frac{p_{\max}}{s} \sum_{i=1}^{s} \text{Var}(o_i(x_i, \tilde{\omega}_i) - o^*) + (1 - p_{\max})\text{Var}(\bar{o}(x_i, \tilde{\omega}_1, \ldots, \tilde{\omega}_s, \tilde{\alpha}) - o^*)$$
$$< \frac{p_{\max}}{s} \sum_{i=1}^{s} \text{Var}(o_i(x_i, \widehat{\omega}_i) - o^*) + (1 - p_{\max})\text{Var}(\bar{o}(x_i, \widehat{\omega}_1, \ldots, \widehat{\omega}_s, \widehat{\alpha}) - o^*). \tag{29}$$

Because $\widehat{\omega}_1, \ldots, \widehat{\omega}_s, \widehat{\alpha}$ minimize the variance of the difference between the combined output and $o^*$, we have

$$\text{Var}(\bar{o}(x_i, \widehat{\omega}_1, \ldots, \widehat{\omega}_s, \widehat{\alpha}) - o^*) < \text{Var}(\bar{o}(x_i, \tilde{\omega}_1, \ldots, \tilde{\omega}_s, \tilde{\alpha}) - o^*). \tag{30}$$

Therefore, we have

$$\sum_{i=1}^{s} \text{Var}(o_i(x_i, \tilde{\omega}_i) - o^*) < \sum_{i=1}^{s} \text{Var}(o_i(x_i, \widehat{\omega}_i) - o^*). \tag{31}$$

$\square$

### A.4 DATASET

**CIFAR-10:** This popular dataset for image classification contains 60,000 32×32 color images, which are divided into 10 distinct classes. The training set is composed of 50,000 images, with each class represented by 5,000 images. The test set includes 10,000 images, with 1,000 images per class.

**CIFAR-100:** This extensive image classification dataset comprises 60,000 32×32 color images, which are categorized into 100 distinct classes. The training set includes 50,000 images, with each class represented by 500 images. The test set has 10,000 images, with 100 images per class.

**ImageNet-16-120:** This dataset, part of the broader ImageNet project, comprises 151,700 16×16 color images distributed across 120 distinct classes. The training set includes 139,200 images, while the validation and test sets consist of 6,000 and 6,500 images, respectively (Chrabaszcz et al., 2017).

**SVHN:** The SVHN dataset is composed of more than 600,000 32×32 color images, each depicting a house number extracted from Google Street View imagery. The dataset is organized into three main parts: a training set with 73,257 images, an extra set containing 531,131 images and a test set comprising 26,032 images. Each image in the dataset features a single digit ranging from 0 to 9, thereby forming a 10-class classification problem (Netzer et al., 2011).

**ImageNet (ILSVRC2012):** This renowned image classification dataset consists of over 1.2 million high-resolution images across 1,000 classes. The training set includes approximately 1.28 million images, while the validation and test sets contain 50,000 and 100,000 images, respectively (Deng et al., 2009).

### A.5 EXPERIMENT SETTINGS

Each experiment involves two stages: architecture search and architecture evaluation. In the search stage, the original dataset is randomly split in half, with one part used for training and the other for validation.

### A.5.1 NAS-BENCH-201 SEARCH SPACE

**Architecture Search:** The NAS-Bench-201 search network (Dong & Yang, 2020) is structured with three stages of cells linked by residual blocks. Each stage consists of five cells, with output channels configured to 16, 32, and 64 for the first, second, and third stages, respectively. The residual blocks double the channels of the input feature map and downsample the spatial dimensions. The candidate operation set $\mathcal{O}$ includes: zero, skip-connection, $1 \times 1$ convolution, $3 \times 3$ convolution, and $3 \times 3$ average pooling.

Network parameters ($\omega$) are optimized using stochastic gradient descent (SGD) with an initial learning rate of 0.025, reduced to 0.001 through cosine annealing. The weight decay is set to 0.0005, and momentum to 0.9. The maximum mutation probability $p_{\max}$ is 0.2. For architecture parameters ($\alpha$), the Adam optimizer is used with a learning rate of $10^{-4}$ and a weight decay rate of 0.001, with momentum terms $\beta_1 = 0.5$ and $\beta_2 = 0.999$. The search on CIFAR-10 runs for 50 epochs.

**Architecture Evaluation:** We utilize the API provided by NAS-Bench-201 (Dong & Yang, 2020) to evaluate the performance of the discovered architectures on the CIFAR-10, CIFAR-100, and ImageNet16-120 datasets.

### A.5.2 DARTS SEARCH SPACE

**Architecture Search:** The DARTS supernet (Liu et al., 2019) consists of normal and reduction cells. These cells form an 8-layer architecture, with reduction cells positioned at layers $N/3$ and $2N/3$ to downsample spatial dimensions and double the channels. The set of candidate operations $\mathcal{O}$ includes $3 \times 3$ and $5 \times 5$ separable and dilated convolutions, $3 \times 3$ max and average pooling, skip-connection, and zero.

Network parameters ($\omega$) are optimized using SGD with an initial learning rate of 0.025, reduced via cosine annealing to 0.001, with 0.0003 weight decay and 0.9 momentum. The maximum mutation probability $p_{\max}$ is 0.125. Architecture parameters ($\alpha$) are optimized with Adam (learning rate $3 \times 10^{-4}$, weight decay 0.001, $\beta_1 = 0.5$, $\beta_2 = 0.999$). The search process spans 50 epochs, using the CIFAR-10 dataset.

**Architecture Evaluation:** For the CIFAR-10 and CIFAR-100 datasets (Xu et al., 2020), the evaluation is conducted on a network consisting of 20 cells, including 18 normal cells and 2 reduction cells. The network starts with 36 channels and is trained for 600 epochs. We use the SGD optimizer with an initial learning rate of 0.025 (cosine decay to 0), momentum of 0.9, weight decay of $3 \times 10^{-4}$, gradient clipping at a norm of 5, and a batch size of 128. Additionally, we incorporate Scheduled-DropPath (with the maximum drop probability linearly increasing to 0.2), cutout (DeVries & Taylor, 2017), and an auxiliary loss with a weight of 0.4.

On the ImageNet dataset (Xu et al., 2020), the evaluative network contains 14 cells (12 normal, 2 reduction) with 48 initial channels. It is trained from scratch for 250 epochs with a batch size of 1024, SGD (momentum of 0.9, learning rate 0.5 with linear decay to 0), weight decay of $3 \times 10^{-5}$, label smoothing, and an auxiliary tower with a weight of 0.4. A learning rate warm-up is applied for the first 5 epochs.

### A.5.3 REDUCED DARTS SEARCH SPACE

**Architecture Search:** The four search spaces defined in R-DARTS (Zela et al., 2020) are subsets of the DARTS search space, except for search space S4, which introduces random noise as one of its operations. Architecture search and evaluation for CIFAR-10, CIFAR-100, and SVHN are performed using settings similar to those in Section 4.2. Depending on the varying needs of different spaces and datasets for preventing performance collapse, we have made targeted adjustments to the setting of $p_{\max}$. Through such adjustments, we ensure that the value of $p_{\max}$ is just right to achieve the effect of suppressing performance collapse, meaning that the number of skip connections in the searched cell architectures will not exceed two. Specific values for the maximum mutation probability $p_{\max}$ are detailed in Table 5.

Table 5: The value of $p_{\max}$ in different datasets and search spaces.

|  | s1 | s2 | s3 | s4 |
|---|---|---|---|---|
| **CIFAR-10** | 0.2 | 0.25 | 0.2 | 0.2 |
| **CIFAR-100** | 0.4 | 0.2 | 0.2 | 0.4 |
| **SVHN** | 0.4 | 0.4 | 0.2 | 0.4 |

**Architecture Evaluation:** On CIFAR-100 and SVHN, architectures are trained from scratch with 16 initial channels and 8 cells, while on CIFAR-10, 36 initial channels and 20 cells are used. All other settings remain consistent with Section 4.2.

### A.5.4 ABLATION STUDY

The same settings as in Section 4.1 are used. When varying the number of training epochs, the mutation probability $p$ starts at 0 and increases linearly to $p_{max} = 0.2$ over the first 50 epochs, after which it remains at $p_{max}$ for searches exceeding 50 epochs.

## A.6 SEARCH COST

One of the critical aspects of NAS is the computational expense associated with searching on large datasets, often quantified in GPU days using the DARTS search space and the CIFAR-10 dataset (Elsken et al., 2019). EM-DARTS uses first-order DARTS, and the edge mutation mechanism adds a step of generating random numbers in each training batch, which has a minor impact on overall computational cost. When an edge undergoes mutation, the unselected operations do not update their weights, thus saving some computational time. We set $p_{max}$ to 0, 0.5, and 1, respectively, to calculate the computational time. The results show that the computational time for different $p_{max}$ values is approximately 0.4 GPU days. Therefore, the total computational cost of our method is 0.4 GPU days on a GTX 1080 Ti GPU. The details are shown in Table 6.

Table 6: Search cost comparison on the DARTS benchmark, provided in GPU days on a 1080 Ti.

| Method | Test Acc (%) | Params (M) | Search Cost (GPU days) |
|---|---|---|---|
| NASNet-A (Zoph et al., 2018) | 97.35 | 3.3 | 2000 |
| ENAS (Pham et al., 2018) | 97.11 | 4.6 | 0.5 |
| DARTS(1st) (Liu et al., 2019) | 97.00 | 3.4 | 0.4 |
| DARTS(2nd) (Liu et al., 2019) | 97.24 | 3.3 | 1.0 |
| SNAS (Xie et al., 2019) | 97.15 | 2.8 | 1.5 |
| GDAS (Dong & Yang, 2019) | 97.07 | 3.4 | 0.3 |
| P-DARTS (Chen et al., 2019) | 97.50 | 3.6 | 0.3 |
| PC-DARTS (Xu et al., 2020) | 97.43 | 3.6 | 0.1 |
| DrNAS (Chen et al., 2021) | 97.46 | 4.0 | 0.4 |
| SDARTS-ADV (Chen & Hsieh, 2020) | 97.39 | 3.3 | 1.3 |
| DARTS- (Chu et al., 2021) | 97.41 | 3.5 | 0.4 |
| DARTS-PT (Wang et al., 2021) | 97.39 | 3.0 | 0.8 |
| Λ-DARTS (Movahedi et al., 2023) | 97.57 | 3.6 | 0.8 |
| **EM-DARTS** | **97.62** | **4.3** | **0.4** |
| **EM-DARTS** ($p_{max} = 0$) | - | - | **0.4** |
| **EM-DARTS** ($p_{max} = 0.5$) | - | - | **0.407** |
| **EM-DARTS** ($p_{max} = 1$) | - | - | **0.394** |

## A.7 SEARCH SPACE CONTAINING ONLY PARAMETRIC OPERATIONS

To validate the performance of EM-DARTS in a search space containing only parametric operations, we defined a new search space S5, in which the set of candidate operations $\mathcal{O}$ includes only $3 \times 3$ and $5 \times 5$ separable and dilated convolutions. The hyperparameter $p_{max}$ for EM-DARTS is set to 0.125. We conducted comparative experiments using DARTS and random sampling as baselines, with all other settings remaining consistent with those in Section 4.2. The experimental results are shown in Table 7. From the table, it can be seen that DARTS outperforms random sampling, indicating that DARTS is indeed an effective search method when it does not experience performance collapse. However, EM-DARTS still achieves the best performance. This suggests that EM-DARTS not only avoids performance collapse but also trains parametric operations more effectively in a search space containing only parametric operations, leading to architectures with better performance.

Table 7: Performance comparison on the search space with only parametric operations, conducted using the CIFAR-10 dataset for both search and evaluation. The reported accuracy values are the mean and standard deviation from four independent runs.

| Method | Random Sampling | DARTS(1st) | EM-DARTS |
|---|---|---|---|
| Test Acc (%) | $96.99 \pm 0.13$ | $97.29 \pm 0.12$ | $97.41 \pm 0.06$ |
| Params (M) | $4.25 \pm 0.14$ | $4.16 \pm 0.15$ | $4.4 \pm 0.25$ |

## A.8 Discovered architectures

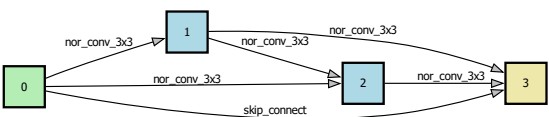

Figure 4: Best searched cell in CIFAR-10 and NAS-Bench-201 benchmark using EM-DARTS.

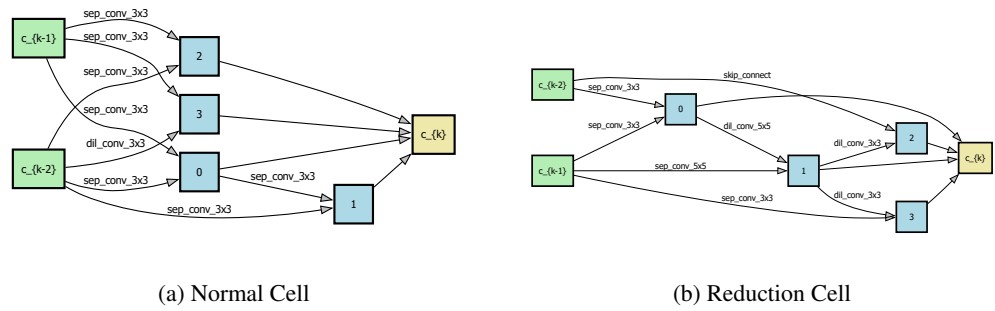

(a) Normal Cell       (b) Reduction Cell

Figure 5: Best searched normal and reduction cells in CIFAR-10 and DARTS benchmark using EM-DARTS.

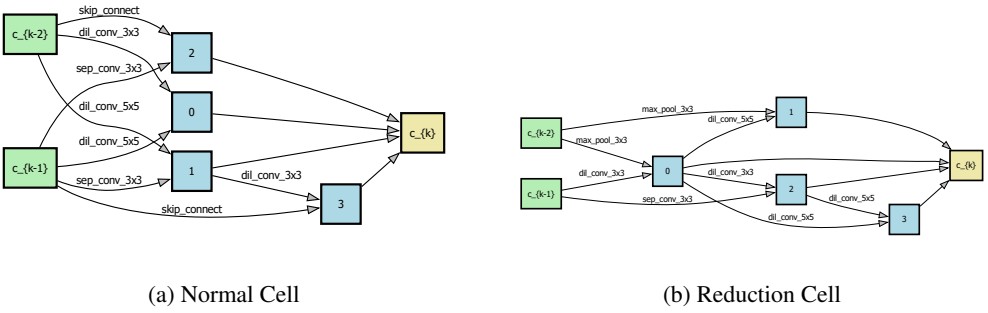

(a) Normal Cell       (b) Reduction Cell

Figure 6: Best searched normal and reduction cells in CIFAR-10 and S1 using EM-DARTS.

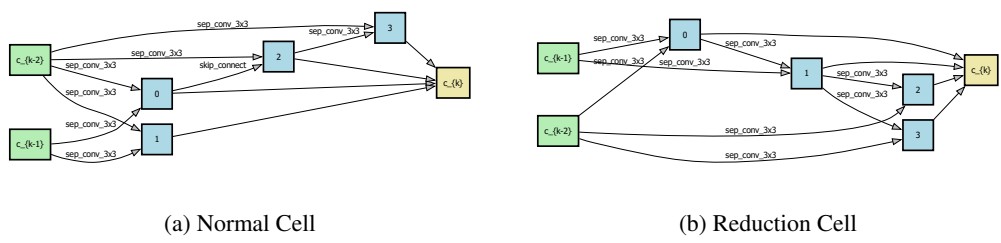

(a) Normal Cell       (b) Reduction Cell

Figure 7: Best searched normal and reduction cells in CIFAR-10 and S2 using EM-DARTS.

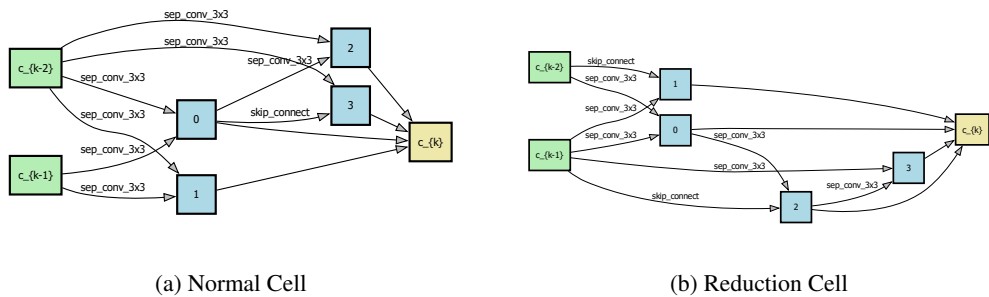

(a) Normal Cell             (b) Reduction Cell

Figure 8: Best searched normal and reduction cells in CIFAR-10 and S3 using EM-DARTS.

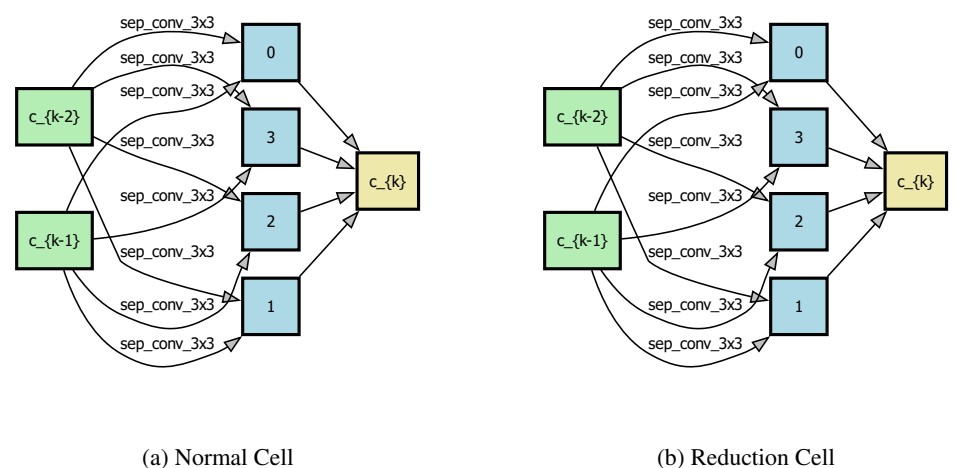

(a) Normal Cell             (b) Reduction Cell

Figure 9: Best searched normal and reduction cells in CIFAR-10 and S4 using EM-DARTS.

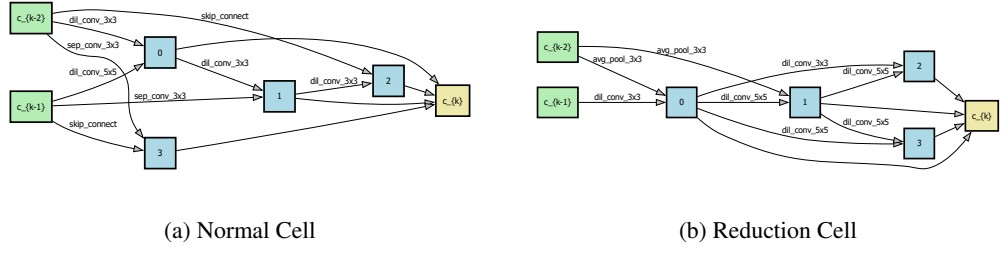

(a) Normal Cell             (b) Reduction Cell

Figure 10: Best searched normal and reduction cells in CIFAR-100 and S1 using EM-DARTS.

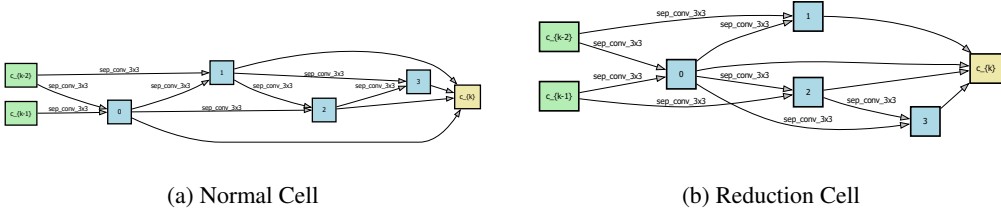

(a) Normal Cell                    (b) Reduction Cell

Figure 11: Best searched normal and reduction cells in CIFAR-100 and S2 using EM-DARTS.

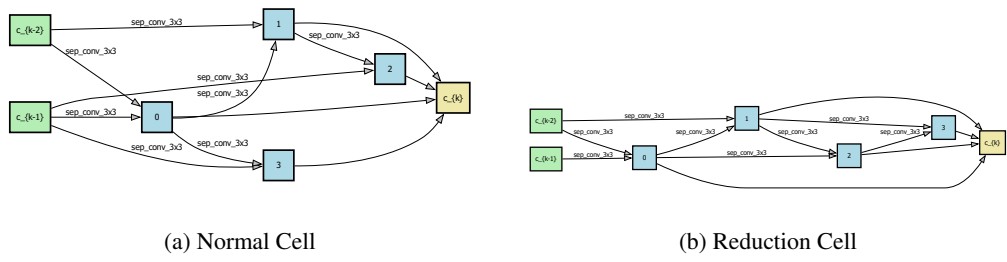

(a) Normal Cell                    (b) Reduction Cell

Figure 12: Best searched normal and reduction cells in CIFAR-100 and S3 using EM-DARTS.

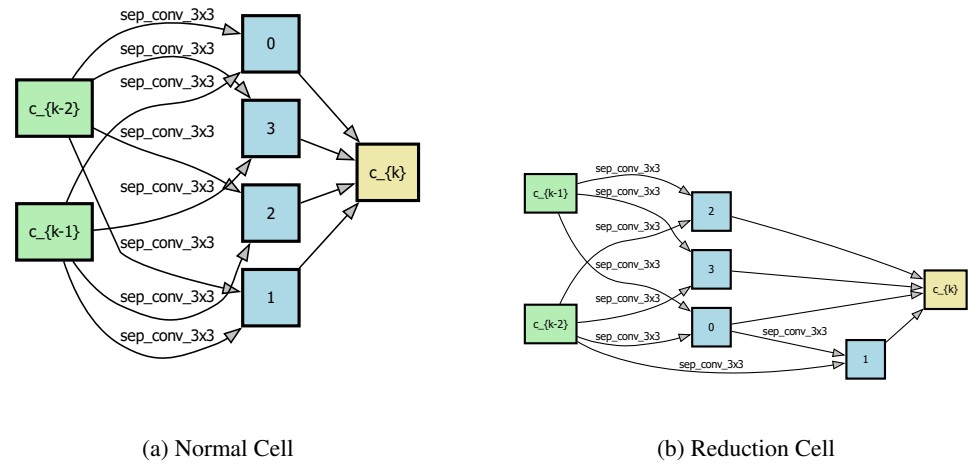

(a) Normal Cell                    (b) Reduction Cell

Figure 13: Best searched normal and reduction cells in CIFAR-100 and S4 using EM-DARTS.

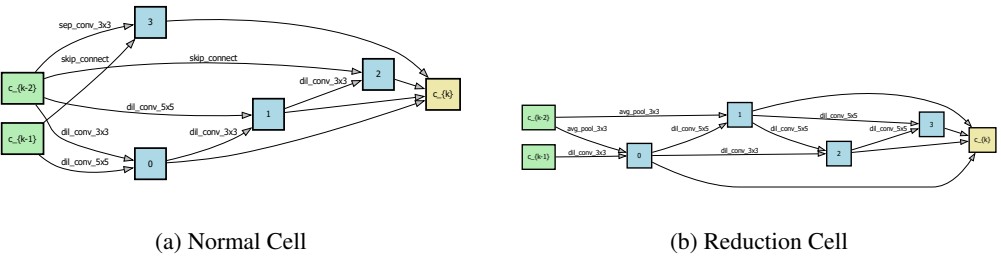

(a) Normal Cell                    (b) Reduction Cell

Figure 14: Best searched normal and reduction cells in SVHN and S1 using EM-DARTS.

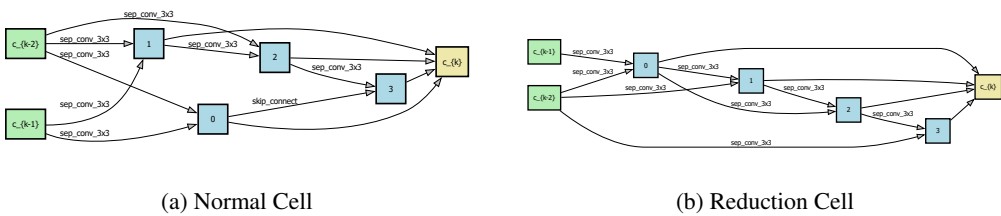

(a) Normal Cell

(b) Reduction Cell

Figure 15: Best searched normal and reduction cells in SVHN and S2 using EM-DARTS.

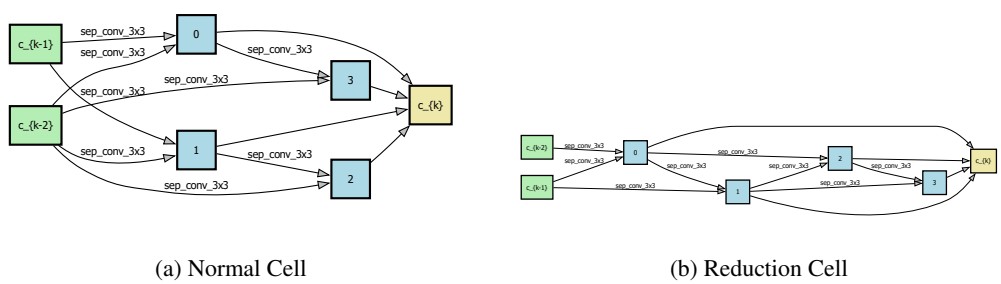

(a) Normal Cell

(b) Reduction Cell

Figure 16: Best searched normal and reduction cells in SVHN and S3 using EM-DARTS.

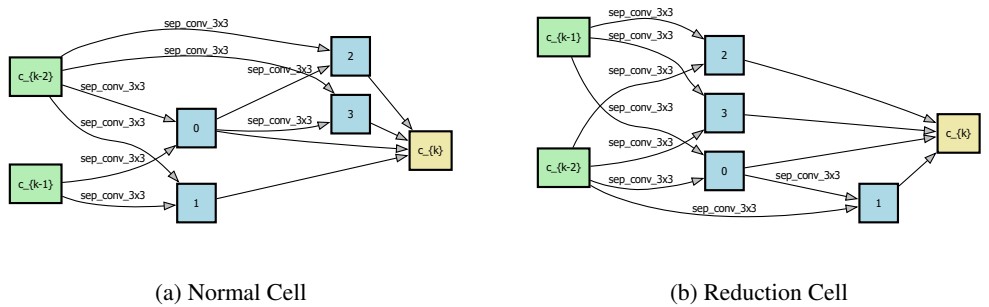

(a) Normal Cell

(b) Reduction Cell

Figure 17: Best searched normal and reduction cells in SVHN and S4 using EM-DARTS.

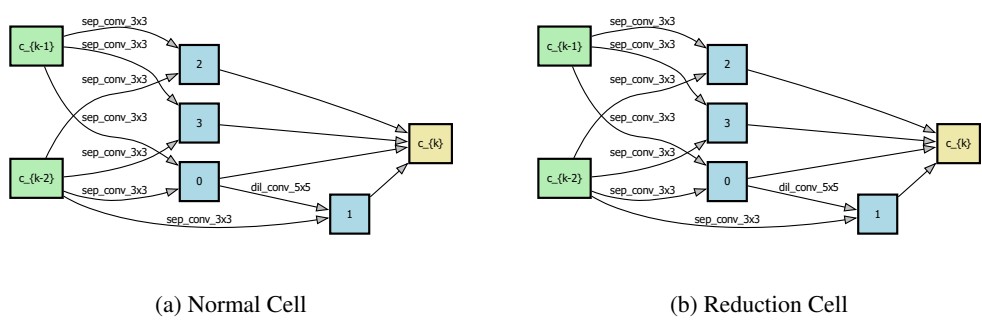

(a) Normal Cell

(b) Reduction Cell

Figure 18: Best searched normal and reduction cells in CIFAR-10 and S5 using EM-DARTS.

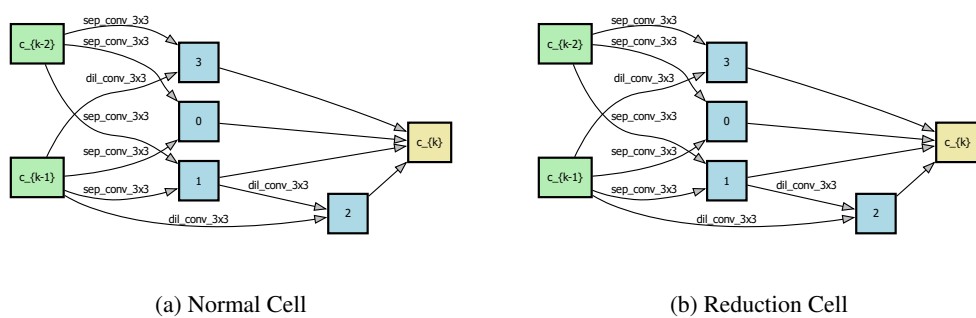

(a) Normal Cell                                    (b) Reduction Cell

Figure 19: Best searched normal and reduction cells in CIFAR-10 and S5 using DARTS.

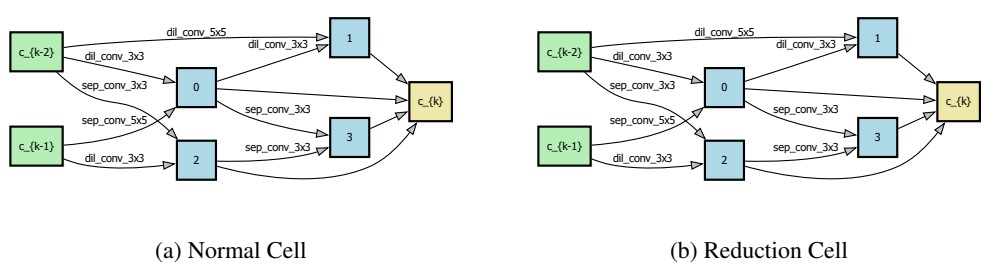

(a) Normal Cell                                    (b) Reduction Cell

Figure 20: Best searched normal and reduction cells in CIFAR-10 and S5 using Random Sampling.

