# OpenReview forum: "EM-DARTS: Preventing Performance Collapse in Differentiable Architecture Search with The Edge Mutation Mechanism"
_ICLR.cc/2025/Conference — Submitted to ICLR 2025_

### Official Review · Reviewer_1Dgb · 2024-11-02

**Soundness:** 3
**Presentation:** 3
**Contribution:** 3
**Rating:** 6
**Confidence:** 3

**Summary:**

This paper introduces EM-DARTS that alleviate the phenomenon of the dominance of skip connections during training of DARTS and the following works. First of all, this study attributes the dominance to two major reasons: 1) the less optimized network parameters due to the coupling of architecture and parametric weights, and 2) the unrolled estimation that all operations attempt to estimate the same feature map leads to a bias towards the choice of skip-connections. An edge mutation differentiable architecture search (EM-DARTS) method is proposed that randomly allows the output of each edge to adopt the one-hot operation. The mutation ratio follows an increasing trend through training to encourage exploitation at an early stage and finally lead to an unbiased estimation. Experimental results show strong performance results on various benchmarks.

**Strengths:**

1. This study is well-motivated as it explains the edge-operation dominance phenomenon well. I find the inverse relationship between the weight of each parametric operation and the variance of the output from its optimal value interesting, and the theoretical derivation is sound to me.
2. The derivation of the biased nature of the unrolled estimation of each operation is also attractive and solid.
3. The experimental performance is competitive when compared with other state-of-the-art neural network search methods.

**Weaknesses:**

1. Although the two convincing reasons for the bias toward skip connection are presented, the solution is still intuitive. The paper does not discuss how mutation can reduce the coupling between parameter optimization and operation search.
2. Given that the mutation can de-bias the optimization target of DARTS, previous studies like DS-NAS that introduced sparsity during DARTS training should also function similarly, despite their different motivations. This paper fails to discuss in detail the comparison with such studies.

**Questions:**

The reviewer appreciates the clear motivation and theoretical analysis, which outweighs the drawbacks. Therefore, a "weak accept" is voted for now.

---

> ### Author Response · Authors · 2024-11-24
> **Replying Weaknesses 1-2**
>
> We appreciate the reviewer's valuable comments. The following are my responses to the questions you raised:
> ### Regarding Weakness 1 (Solution Still Intuitive, Lack of Discussion on How Mutation Reduces Coupling)
>
> We understand the reviewer's concern about the intuitiveness of the solution and have revised the paper to provide a more detailed discussion on how the edge mutation mechanism reduces the coupling between the weights of parametric operations and the architecture parameters.
>
> In fact, the edge mutation mechanism introduces randomness during the edge weight update process in the supernet, allowing parametric operations to independently receive input data and perform forward and backward propagation. This reduces the influence of architecture parameters on parametric operations, thereby decreasing the coupling between the weights of parametric operations and the architecture parameters.
>
> ### Regarding Weakness 2 (Comparison with Methods Introducing Sparsity, Such as DS-NAS)
>
> We have revised the paper to include a comparison with methods that introduce sparsity, such as DS-NAS, in the related work section.
>
> - **SPOS**: Reduces computational resource usage by randomly selecting a single path for training within a supernet, maintaining the diversity of the search space and enhancing search efficiency.
> - **GDAS**: Builds on SPOS by using the Gumbel-Softmax distribution to make the search space continuous, allowing for gradient-based optimization of architecture parameters, thus further improving efficiency and stability.
> - **DSNAS**: Adds sparsity constraints to SPOS, directly optimizing network architectures and eliminating the need for parameter retraining, simplifying the search process.
>
> Inspired by the sparse strategy of DSNAS, which can reduce the coupling between architectural parameters and network weights, we propose EM-DARTS. However, unlike the aforementioned methods that primarily focus on improving search efficiency, EM-DARTS is specifically designed to address the issue of performance collapse in DARTS. By introducing an edge mutation mechanism, EM-DARTS aims to mitigate the adverse effects of the continuous relaxation framework on the training of parametric operations. This edge mutation mechanism can be seen as an extension of sparsity constraints. Because the purpose of using edge mutation is to enhance the training level of parametric operations, it differs from the sparsity strategies in SPOS and DSNAS. In EM-DARTS, edge mutations are restricted to parametric operations, rather than all operations. Moreover, EM-DARTS does not enforce every edge to mutate into a single operation; instead, each edge mutates into a specific parametric operation with a certain probability $p$.

---

> > ### Comment · Reviewer_1Dgb · 2024-11-27
> > **A follow-up question**
> >
> > I would like to thank the authors for the effort to clarify my concerns. The authors view the major difference between EM-DARTS and the aforementioned sparsity-involving methods as the mutation on only parametric operations. However, this is unclear to me as the impact of edge mutation on non-parametric operations is not shown in the paper. If the edge mutation on all operations does not significantly influence, what explains the performance gain over methods that have introduced sparsity?

---

> ### Author Response · Authors · 2024-11-27
> **Replying the follow-up question**
>
> Thank you for your valuable feedback, which has greatly assisted us in clarifying and enhancing our explanation of EM-DARTS. We have already supplemented the purpose of using edge mutation, which is to enhance the training level of parametric operations, in the related work section.
>
> First, the primary distinction between parametric and non-parametric operations lies in the fact that parametric operations require their weights to be updated through training, such as depthwise separable convolutions, dilated convolutions, and standard convolutions; whereas the weights of non-parametric operations are pre-determined and remain fixed, like max pooling, average pooling, and residual connections. Therefore, whether or not an edge mutation strategy is adopted, it does not impact the weights of non-parametric operations. This is also the reason why we did not delve into the effects of edge mutation on non-parametric operations in our paper.
>
> Second, based on the analysis from Proposition 1 and Proposition 2, we observe that in the continuous relaxation framework, parametric operations are susceptible to the influence of architectural parameters, which often leads to insufficient training. Therefore, when the DARTS method converges, the architectural parameters of parametric operations actually reflect their relative importance under conditions of inadequate training. This indicates that within the DARTS framework, architectural parameters fail to accurately represent the true importance of corresponding operations, as the importance of parametric operations is often underestimated.
>
> Next, the purpose of introducing sparsity in the process of updating network weights in EM-DARTS is to enhance the training level of parametric operations. The effectiveness of this mechanism has been verified in Theorem 1. Through such improvements, we can mitigate the issue of parametric operations being underestimated in DARTS, thereby enhancing the accuracy of architectural parameters.  Therefore, compared to other methods, our approach can achieve significant performance enhancements.

---

### Official Review · Reviewer_QBxG · 2024-11-03

**Soundness:** 2
**Presentation:** 2
**Contribution:** 2
**Rating:** 3
**Confidence:** 3

**Summary:**

This paper proposes an edge mutation approach to improve the robustness of DARTS.

**Strengths:**

+ The motivation of the paper is clear. It is well-known that DARTS has many robustness issues and there have been various ways to address that.
+ The observation that insufficient training may be the cause for some of the issues of DARTS makes sense.

**Weaknesses:**

- The technical contributions seem to be incremental. Essentially it is a form of doing DARTS and SPOS together.
- The theoretical analysis is a bit misleading, and it seems to be very close to what has been discussed in existing work like DARTS-PT.
- The results are also less convincing. For instance, on DARTS space, the discovered models are significantly larger than the competing approaches, which probably is the main reason of performance increase. Also the search cost reported is the same with DARTS, but why?

**Questions:**

* What if you only do the mutation at different stages of training? Would that make major differences?
* Why your search cost is the same with DARTS when you have to do extra work of mutation?

---

> ### Author Response · Authors · 2024-11-24
> **Replying  Weaknesses 1-2**
>
> We appreciate the reviewer's valuable comments. The following are my responses to the questions you raised:
>
> ### Regarding Weakness 1 (Technical Contributions Seem Incremental, Essentially a Combination of DARTS and SPOS)
>
> Our approach is not merely a combination of DARTS and SPOS but introduces significant distinctions and innovations. We have re-examined and addressed the issue of performance collapse in DARTS from an entirely new perspective, an area unexplored by predecessors. Compared to DSNAS, which tends to be a direct combination of DARTS and SPOS, our method demonstrates clear differences and has been empirically shown to outperform DSNAS, as detailed in Table 1 of the main text.
>
> Specifically:
>
> - **SPOS**: By randomly selecting a single path within the supernet for training, it effectively reduces the consumption of computational resources while maintaining the diversity of the search space and enhancing search efficiency.
> - **GDAS**: Building on SPOS, it adopts the Gumbel-Softmax distribution to make the search space continuous, allowing architecture parameters to be optimized through gradient descent, thereby further improving the efficiency and stability of the search process.
> - **DSNAS**: It adds sparsity constraints to SPOS, simplifying the optimization process of network architectures and eliminating the need for parameter retraining.
>
> Inspired by the sparse strategy of DSNAS, which can reduce the coupling between architectural parameters and network weights, we propose EM-DARTS. However, unlike the aforementioned methods that primarily focus on improving search efficiency, EM-DARTS is specifically designed to address the issue of performance collapse in DARTS. By introducing an edge mutation mechanism, EM-DARTS aims to mitigate the adverse effects of the continuous relaxation framework on the training of parametric operations. This edge mutation mechanism can be seen as an extension of sparsity constraints. Because the purpose of using edge mutation is to enhance the training level of parametric operations, it differs from the sparsity strategies in SPOS and DSNAS. In EM-DARTS, edge mutations are restricted to parametric operations, rather than all operations. Moreover, EM-DARTS does not enforce every edge to mutate into a single operation; instead, each edge mutates into a specific parametric operation with a certain probability $ p $.
>
> ### Regarding Weakness 2 (Misleading and Similarity of Theoretical Analysis):
> Our theoretical section, including Proposition 1, Proposition 2, and Theorem 1, provides detailed proofs in the appendix. We speculate that the reviewer may have some misunderstandings regarding Proposition 1, and thus we would like to take this opportunity to clarify any related confusion. While our theoretical analysis was indeed inspired by certain concepts in DARTS-PT, we have conducted in-depth research and innovation based on these foundations, ensuring that our conclusions and analyses offer unique contributions. Below are specific explanations of these unique contributions:
>
> 1. **Inspiration and Innovation**:
>    - **Inspiration from DARTS-PT**: We did indeed draw upon some theoretical frameworks from DARTS-PT, particularly its analysis of performance collapse. DARTS-PT highlights issues with existing evaluation systems, especially the inability of architecture parameters to accurately reflect the importance of operations.
>    - **Unique Contributions**: However, our work extends DARTS-PT's research further. We discovered that the fundamental reason for architecture parameters failing to accurately reflect the importance of operations is the undertraining of parametric operations, and we support our findings with Proposition 2. Specifically, we prove that inadequate training of parametric operations under the continuous relaxation framework is the primary cause of performance collapse.
>
> 2. **Theoretical Consistency**:
>    - **Proposition 1 of DARTS-PT**: Proposition 1 of DARTS-PT states that architecture parameters cannot accurately represent the importance of operations, leading them to abandon the use of architecture parameters to find the optimal architecture.
>    - **Proposition 1 of EM-DARTS**: Our Proposition 1 aligns with DARTS-PT's Proposition 1 in the case of two candidate operations. Specifically, the conclusion of DARTS-PT's Proposition 1 is equivalent to our Proposition 1 when there are only two candidate operations. These two propositions are not contradictory but complementary.
>
> 3. **Detailed Derivations**:
>    - **Consistency Proof**: Below is the derivation of our Proposition 1 based on DARTS-PT's Proposition 1. The only additional steps are equations (1) and (2); otherwise, the computational steps are identical to those in the proof of DARTS-PT's Proposition 1.

---

> ### Author Response · Authors · 2024-11-24
> **Proof of Weaknesses2**
>
> ### Proof
>
> Let $\theta_{skip} = \text{Softmax}(\alpha_{skip})$ and $\theta_{conv} = \text{Softmax}(\alpha_{conv})$.
>
> Then the mixed operation can be written as:
> \begin{align}
> \\overline{m}\_e(x\_e) = \\theta_{conv} o\_e(x_e) + \\theta\_{skip} x\_e.
> \end{align}
> We formally formulate the objective to be:
>
> \begin{align}
> 		\\min\_{\\theta\_{skip}, \\theta\_{conv}} & \\text{Var}(\\overline{m}\_e(x\_e) - m\^*) \\\\
> 			s.t. & \\quad \\theta\_{skip} + \\theta\_{conv} = 1
> \end{align}
>
> This constrained optimization problem can be solved using Lagrangian multipliers:
>
> $L(\\theta\_{skip},\\theta\_{conv},\\lambda)\$
>
> $=\\text{Var}(\\overline{m}\_e(x\_e)-m\^*) -\\lambda (\\theta_{skip}+\\theta_{conv} - 1)$
>
> $= \text{Var}(\theta_{conv} o_e(x_e) + \theta_{skip} x_e - m^*) - \lambda (\theta_{skip} + \theta_{conv} - 1)$
>
> $= \text{Var}(\theta_{conv} (o_e(x_e) - m^*)  + \theta_{skip} (x_e - m^*)) - \lambda (\theta_{skip} + \theta_{conv} - 1)$
>
> $= \theta_{conv}^2 \text{Var}(o_e(x_e) - m^*) + \theta_{skip}^2 \text{Var}(x_e - m^*)$
>
> $\quad + 2\theta_{conv}\theta_{skip} \text{Cov}(o_e(x_e) - m^*, x_e - m^*) - \lambda (\theta_{skip} + \theta_{conv} - 1)$
>
> Taking partial derivatives and setting them to zero:
> \begin{align}
> \frac{\partial L}{\partial \lambda} &=  \theta_{conv} + \theta_{skip} - 1 = 0\\\\
> \frac{\partial L}{\partial \theta_{conv}} &= 2\theta_{conv}\text{Var}(o_e(x_e) - m^*) + 2\theta_{skip}\text{Cov}(o_e(x_e) - m^*, x_e - m^*)\nonumber\\\\
> &\quad - \lambda = 0 \\\\
> \frac{\partial L}{\partial \theta_{skip}} &= 2\theta_{conv}\text{Cov}(o_e(x_e) - m^*, x_e - m^*) + 2\theta_{skip}\text{Var}(x_e - m^*)\nonumber\\\\
> &\quad - \lambda = 0
> \end{align}
>
> Solving these equations gives:
> \begin{align}
> \theta_{conv}^* &= \frac{\text{Var}(x_e - m^*) - \text{Cov}(o_e(x_e) - m^*, x_e - m^*)}{ \text{Var}(o_e(x_e) - m^*) - 2\text{Cov}(o_e(x_e) - m^*, x_e - m^*) + \text{Var}(x_e - m^*)}\\\\
> \theta_{skip}^* &= \frac{\text{Var}(o_e(x_e) - m^*) - \text{Cov}(o_e(x_e) - m^*, x_e - m^*)}{ \text{Var}(o_e(x_e) - m^*) - 2\text{Cov}(o_e(x_e) - m^*, x_e - m^*) + \text{Var}(x_e - m^*)}
> \end{align}
>
> Dividing the numerator and denominator by $\text{Var}(x_e - m^*) \text{Var}(o_e(x_e) - m^*)$:
> \begin{align}
> \theta_{conv}^* &= \frac{ \frac{1}{ \text{Var}(o_e(x_e) - m^*)} - \frac{\text{Cov}(o_e(x_e) - m^*, x_e - m^*)}{ \text{Var}(x_e - m^*) \text{Var}(o_e(x_e) - m^*)}}
> {\frac{1}{ \text{Var}(x_e - m^*)} - \frac{2\text{Cov}(o_e(x_e) - m^*, x_e - m^*)}{ \text{Var}(x_e - m^*) \text{Var}(o_e(x_e) - m^*)} + \frac{1}{ \text{Var}(o_e(x_e) - m^*)}} \\qquad  \\qquad  (1)\\\\
> \theta_{skip}^* &= \frac{\frac{1}{ \text{Var}(x_e - m^*)} - \frac{\text{Cov}(o_e(x_e) - m^*, x_e - m^*)}{ \text{Var}(x_e - m^*) \text{Var}(o_e(x_e) - m^*)}}
> {\frac{1}{ \text{Var}(x_e - m^*)} - \frac{2\text{Cov}(o_e(x_e) - m^*, x_e - m^*)}{ \text{Var}(x_e - m^*) \text{Var}(o_e(x_e) - m^*)} + \frac{1}{ \text{Var}(o_e(x_e) - m^*)}}  \\qquad  \\qquad  (2)
> \end{align}
>
> Aligning with DARTS, we get:
> \begin{align}
> \alpha_{conv}^* &= \log \left[ \frac{1}{ \text{Var}(o_e(x_e) - m^*)} - \frac{\text{Cov}(o_e(x_e) - m^*, x_e - m^*)}{ \text{Var}(x_e - m^*) \text{Var}(o_e(x_e) - m^*)} \right] + C\\\\
> \alpha_{skip}^* &= \log \left[ \frac{1}{ \text{Var}(x_e - m^*)} - \frac{\text{Cov}(o_e(x_e) - m^*, x_e - m^*)}{ \text{Var}(x_e - m^*) \text{Var}(o_e(x_e) - m^*)} \right] + C
> \end{align}
>
> The key difference between $\alpha_{skip}$ and $\alpha_{conv}$ lies in the first term inside the logarithm, thus:
> \begin{align}
> \alpha_{conv}^* &\propto \frac{1}{ \text{Var}(o_e(x_e) - m^*)} \\\\
> \alpha_{skip}^* &\propto \frac{1}{ \text{Var}(x_e - m^*)}
> \end{align}

---

> ### Author Response · Authors · 2024-11-24
> **Replying Weaknesses 3 and Question 1-2**
>
> ### Regarding Weakness 3 (Performance Improvement Due to Increased Model Size)
>
> In fact, the task of DARTS involves selecting the optimal architecture within a given search space. When evaluating different architectures, we ensure that the initial channel numbers, layer counts, and other conditions of all evaluated networks remain consistent. Therefore, the improvement in performance should not be simply attributed to an increase in model size. To further validate this point, we conducted experiments on the CIFAR-10 dataset using the architecture with the largest number of parameters within the DARTS search space. This architecture contains 5.1M parameters and is structured as follows:
>
> ```
> Genotype(
>     normal=[('sep_conv_5x5', 1), ('sep_conv_5x5', 0), ('sep_conv_5x5', 2), ('sep_conv_5x5', 0), ('sep_conv_5x5', 3),
>             ('sep_conv_5x5', 1), ('sep_conv_5x5', 2), ('sep_conv_5x5', 1)],
>     normal_concat=range(2, 6),
>     reduce=[('sep_conv_5x5', 0), ('sep_conv_5x5', 1), ('sep_conv_5x5', 0), ('sep_conv_5x5', 2), ('sep_conv_5x5', 3),
>             ('sep_conv_5x5', 2), ('sep_conv_5x5', 4), ('sep_conv_5x5', 3)],
>     reduce_concat=range(2, 6)
> )
> ```
>
> The experimental results show that the test accuracy of this architecture is only 96.87%, and experimental records can be found in the `result` folder of the attachment. This further demonstrates that the performance improvement is not simply due to an increase in model size.
>
> ### Regarding Question 1 (Mutation at Different Training Stages)
> We set up three phases of edge mutation: an early phase with a mutation probability of 0.2 for the first 1/3 of epochs, a middle phase with a mutation probability of 0.2 for the next 1/3 of epochs, and a late phase with a mutation probability of 0.2 for the last 1/3 of epochs. We plotted the results of the early, middle, and late phases in Figure 2 of the main text. From the figure, it can be observed that the architectures found during the mutation periods in all three phases perform well, but the architectures vary over time, and the performance remains good until the end of the search. Interestingly, all methods that use the edge mutation mechanism successfully avoid performance collapse, with final architectures achieving accuracies exceeding 90%, highlighting the effectiveness of the mutation mechanism. However, the performance of the architectures obtained using these methods is still inferior to those obtained with a mutation probability that increases linearly from 0 to 0.2. We find that the increasing probability strategy ensures that the optimal architecture is found early and remains stable. These findings emphasize the necessity of the increasing probability strategy.
>
> **Table:** Under the NAS-Bench-201 framework, the accuracy (%) results obtained from experiments on the CIFAR-10 dataset with different strategies.
> |  strategy | Exp1 | Exp2 | Exp3 | Exp4 | Mean|
> |--------------|------|------|------|------|------|
> | **DARTS** | 39.77 | 39.77 | 39.77 | 39.77 | 39.77 ± 0 |
> | **p=0.1** | 90.40 | 90.57 | 91.5 | 91.12 |90.90 ± 0.44|
> | **p=0.2** | 90.57 | 90.57 | 90.4 | 90.32 |90.47 ± 0.11|
> | **early** | 91.12 | 90.12 | 89.81 | 90.5 | 90.39 ± 0.49|
> | **middle** | 91.12 | 91.12 | 90.51 | 91.50 | 91.06 ± 0.35|
> | **late** | 91.12 | 91.55 | 91.55 | 91.50 | 91.43 ± 0.18|
> | **linear** | 91.55 | 91.55 | 91.55 | 91.55 | 91.55 ± 0|
>
> ### Regarding Questions 2 (Search Cost):
>
> EM-DARTS uses first-order DARTS, and the edge mutation mechanism adds a step of generating random numbers in each training batch, which has a minor impact on overall computational cost. When an edge undergoes mutation, the unselected operations do not update their weights, thus saving some computational time. We set $p_{\max}$ to 0, 0.5, and 1, respectively, to calculate the computational time. The results show that the computational time for different $p_{\max}$ values is approximately 0.4 GPU days. Therefore, the total computational cost of our method is 0.4 GPU days on a GTX 1080 Ti GPU.
>
>
> | Method                         | Computational Cost (GPU days) |
> |---------------------------------|-------------------------------|
> | **DARTS**                    | **0.4**                       |
> | **EM-DARTS ($p_{\max} = 0$)**     | **0.4**                       |
> | **EM-DARTS ($p_{\max} = 0.5$)**    | **0.407**                     |
> | **EM-DARTS ($p_{\max} = 1$)**   | **0.394**                     |

---

### Official Review · Reviewer_8UyK · 2024-11-03

**Soundness:** 2
**Presentation:** 2
**Contribution:** 2
**Rating:** 6
**Confidence:** 4

**Summary:**

This paper focuses on mitigating the failure mode of the DARTS method which causes it to select architectures which are dominated by parameterless skip connection operations. The authors argue that while several explanations for this failure mode have been posited, ranging from overfitting during the search phase of DARTS to the unfair advantage of skip-connections in the optimization process, they all overlook the effect of the continuous relaxation of the search space on the parametric operations. They prove, theoretically, that the continuous relaxation framework causes the parametric operations to learn not the optimal features for themselves, but features that contribute to the overall performance of the edge. This finding intuitively explains the discretization gap to some degree. The authors suggest modifying the method used to combine the feature maps from operations on a given edge to generate the output feature map for that edge. Typically, the output feature map of an edge is produced by summing the feature maps of the operations on that edge, each weighted by their softmax-normalized architectural weights. The authors suggest a new approach: sample either this output feature map or the output of a randomly sampled parametrized operation as the output of the edge. This strategy encourages the parametrized operations to train more robustly on their own, rather than simply complementing the other operations on the edge.

**Strengths:**

### Originality

The proposed method can be seen as a hybrid of DARTS and other methods such as GDAS [1] which trains the supernet along one randomly sampled path at a time. However, the motivation for this approach is theoretically justified.

### Quality
The paper provides a decent ablation study of the main components of their method, albeit on a small tabular benchmark (NAS-Bench-201). These ablations show that (1) randomly sampling either the mixed feature map or the output feature map of a parametrized operation can induce stability to the training phase of DARTS and (2) this stability holds for an extended period of training the supernet (up to 400 epochs).

### Clarity
The writing is mostly clear and follows a neat narrative structure. The method is well-motivated and easy to understand.

### Significance
The main significance of the paper lies in its contribution to the theoretical understanding of the failure mode of DARTS.
Specifically, it shows that the continuous relaxation scheme of the supernet in DARTS does not allow the parametrized operations to learn the representations which are as close as possible to the optimal representations for a given edge.

**Weaknesses:**

The main weakness of the paper is the evaluation pipeline chosen for the DARTS architectures. It is mentioned in Section A.5.2 that "due to introducing more parametrized operations, the training is extended from 600 to 800 epochs". This significant deviation from the DARTS evaluation pipeline makes the results of the experiments unreliable. In my experience, the test performance of DARTS models do not plateau at 600 epochs of training. It is conceivable that the performance of the models could simply be an artifact of training the models 33% longer. The argument that the model is trained longer due to a higher number of parameters is also not strong, considering that DrNAS, which discovers models with 4M parameters (compared to EM-DARTS with an average of 4.3M), also evaluates the model with 600 epochs of training. Regrettably, this renders the comparisons in Table 2 both unfair and invalid.

The paper does not explicitly state the number of epochs used in training the Reduced DARTS models. It simply mentions that it is the same as DARTS in Section 4.2. If these models have also been trained for longer, then the comparisons in Table 3 are not fair either.

The experiments on NAS-Bench-201 look robust and fair, since they are all evaluated with a tabular benchmark. However, the results on this benchmark alone do not adequately defend the proposed method.

A few other minor issues are:
1. Grammatical errors in the text. E.g., "we analyzes" in the abstract in L017.
2. The text in Figure 1 is not clearly legible.
3. Incorrect style of citation in L196.

**Questions:**

1. Can the authors provide the results of EM-DARTS models trained for 600 epochs?
2. What is the motivation for picking 800 epochs, specifically? Why not 700, for example, or 900?
3. Have you tracked the trajectories of the number of parameterless operations in the discretized models as the training of the supernet progresses? Empirically, is it not possible that the method simply biases the optimization in favour of architectures with more parametrized operations? As seen in Figure 5, the normal cells have no parameterless operations, while the reduction cell has only one parameterless operation (a skip connection) in its eight edges.
4. How does EM-DARTS perform against a random sampling baseline, where only parameterized operations are included in the search space?

---

> ### Author Response · Authors · 2024-11-24
> **Replying minor issues 1-3 and Questions 1 and 2**
>
> We appreciate the reviewer's valuable comments. The following are my responses to the questions you raised:
>
> ### Regarding Grammar Errors in the Text:
> For example, "we analyzes" in L017 of the abstract.
> We have corrected the grammar error in the abstract, changing "we analyzes" to "we analyze."
>
> ### Regarding Text Clarity in Figure 1:
> The text in Figure 1 was not clear enough.
> We have remade Figure 1 to ensure that all text is clear and legible.
>
> ### Regarding Incorrect Citation Format at L196:
> We have corrected the content at L196, and we have also fixed similar citation errors throughout the document.
>
> ### Regarding Questions 1 and 2 (Rationale for Choosing 800 Epochs)
>
> The reviewer pointed out that we chose 800 epochs for training and questioned the rationale behind this choice. Our initial motivation for selecting 800 epochs was based on the following considerations:
>
> 1. **Impact of More Skip Connections**: In preliminary experiments, we observed that architectures obtained from other DARTS improvement methods contained more skip connection operations, making the models easier to train fully. Therefore, we believed that extending the training duration could help our model converge better.
> 2. **Experimental Exploration**: The choice of 800 epochs was initially a randomly selected value during the exploratory phase. Our goal was to ensure that the model received sufficient training by using a longer training period.
>
> To ensure fairness and comparability in our experiments, we have re-conducted all experiments with a unified training duration of 600 epochs. The results are as follows. The latest experimental results have been updated in the paper to ensure that all methods are evaluated under the same training conditions. Experimental records can be found in the `result` folder of the attachment. Below are the results we obtained from training and evaluating the models for 600 epochs:
>
> **Table:** Under the DARTS framework, the accuracy (%) results obtained from experiments on the CIFAR-10, CIFAR-100, and ImageNet datasets.
>
> | Dataset      | Exp1  | Exp2  | Exp3  | Exp4  |
> |--------------|-------|-------|-------|-------|
> | **CIFAR-10** | 97.64 | 97.57 | 97.59 | 97.67 |
> | **CIFAR-100**| 84.05 | 83.78 | 83.83 | 84.19 |
> | **ImageNet** | -     | -     | -     | 76.2  |
>
> **Table:** Under the Reduced DARTS framework, the test error rate (%) results obtained from experiments on the CIFAR-10, CIFAR-100, and SVHN datasets.
>
> | Dataset      | s1    | s2    | s3    | s4    |
> |--------------|-------|-------|-------|-------|
> | **CIFAR-10** | 2.74  | 2.39  | 2.44  | 2.32  |
> | **CIFAR-100**| 23.27 | 21.47 | 20.33 | 20.35 |
> | **SVHN**     | 2.49  | 2.34  | 2.30  | 2.27  |
>
> Explanation for why 600 epochs did not lead to a significant decrease in evaluation network performance:
>
> 1. **Impact of Learning Rate**: The performance of the evaluation network is significantly influenced by the learning rate. From our experimental records, we observed that there is a long period during which the model's accuracy does not improve further. Only when the learning rate drops below a certain threshold does the model's accuracy continue to increase.
>
> 2. **Error in Data Augmentation Technique (Cutout)**: There was an error in the original code regarding the use of the Cutout data augmentation technique. Initially, the probability of applying Cutout increased gradually from 0 to 1 over the epochs. However, we noticed that other methods typically set the Cutout probability directly to 1. To unify the hyperparameter settings, we adjusted the Cutout probability to always be 1. We found that the previous setting led to overfitting around 500 epochs, hindering further performance improvement. After aligning with the settings used by other methods, the overfitting issue was significantly mitigated.

---

> ### Author Response · Authors · 2024-11-24
> **Replying Questions 3 and Questions 4**
>
> ### Regarding Question 3 (Preference for Parametric vs. Non-Parametric Operations)
>
> The reviewer asked whether the edge mutation mechanism biases the selection toward more parametric operations. Here are our clarifications:
>
> 1. **Difference Between Parametric and Non-Parametric Operations**: We categorize operations into parametric and non-parametric operations primarily because parametric operations have weights that need to be optimized through training, whereas non-parametric operations have fixed weights that do not require training. Since non-parametric operations do not need training, the edge mutation mechanism, which introduces randomness during network weight updates, does not affect the weights of non-parametric operations.
>
> 2. **Edge Mutation Mechanism**: The edge mutation mechanism introduces randomness only during the network weight update process, aiming to improve the training quality of parametric operations. During the architecture parameter update stage, the edge mutation mechanism does not play a role, and the update of architecture parameters is still based on the performance of the current operations. Therefore, the edge mutation mechanism does not influence the training of architecture parameters.
>
> 3. **Experimental Results**: The final architecture contains more parametric operations than non-parametric operations. This is because the edge mutation mechanism indeed improves the training quality of parametric operations, making their architecture parameters larger and thus more likely to be selected in the final architecture. In fact, the architectural parameters in EM-DARTS are more accurate in reflecting the actual importance of all operations compared to those in DARTS.
>
> ### Regarding Questions 4 (Comparison with Random Sampling Baseline)
>
> The reviewer mentioned whether we compared our method with a random sampling baseline, particularly in a search space containing only parametric operations. We have conducted the corresponding experiments and added the results to the appendix of the paper:
>
> To validate the performance of EM-DARTS in a search space containing only parametric operations, we defined a new search space  S5, where the set of candidate operations $ \mathcal{O} $ includes only $3\times3$ and $ 5\times5 $ separable and dilated convolutions. For EM-DARTS, the hyperparameter $ p_{\max}$ was set to 0.125. We conducted comparative experiments using DARTS and random sampling as baselines, with all other settings remaining consistent with those in DARTS search space.
>
> From the table, it is evident that DARTS outperforms random sampling, indicating that DARTS is indeed an effective search method when it does not experience performance collapse. However, EM-DARTS still achieves the best performance. This suggests that EM-DARTS not only avoids performance collapse but also trains parametric operations more effectively in a search space containing only parametric operations, resulting in better-performing architectures.
>
> **Table:** Under S5, the accuracy (%) results obtained from experiments on the CIFAR-10 datasets.
> |   Method   | Exp1 | Exp2                | Exp3         | Exp4 | mean |
> |--------------|------|---------------------|--------------|------| --------|
> | **Random Sampling** | 96.9 | 97.09 | 96.85 | 97.11 | 96.99 ± 0.13 |
> | **DARTS(1st)** | 97.33 | 97.36 | 97.37 | 97.11 | 97.29 ± 0.12|
> | **EM-DARTS** | 97.44 | 97.35 | 97.49 | 97.40| **97.41 ± 0.06**  |
>
>  **Regarding the performance of EM-DARTS in S5 being inferior to that in the DARTS search space, this can be understood from the following points:**
>
> Firstly, the S5 search space is a subset of the DARTS search space. Models found in a smaller search space typically do not outperform models discovered in a larger search space, which aligns with intuition and practical experience.
>
> Secondly, the key hyperparameter $p_{\max}$ in our designed edge mutation mechanism was chosen based on its effectiveness in suppressing performance collapse phenomena. However, in the S5 search space, due to the absence of skip connection operations, such performance collapse issues theoretically do not exist. As a result, the setting of $p_{\max}$ loses its reference point, making the fine-tuning process extremely challenging.
>
> The setting of $p_{\max} = 0.125$ is derived from the best practices obtained through experiments in the DARTS search space and may not be entirely suitable when directly applied to the S5 search space. Given the time constraints of the research, we were unable to further optimize the value of $p_{\max}$ for the S5 search space. Despite these conditions, EM-DARTS has still demonstrated superior performance compared to other methods, indicating that the approach possesses strong advantages and adaptability. In future research, we will consider conducting more detailed adjustments of hyperparameters for different search spaces to achieve even better outcomes.

---

> > ### Comment · Reviewer_8UyK · 2024-11-26
> >
> > I thank the authors for their response.
> >
> > My concern regarding the evaluation pipeline, which informed my rating for soundness of the paper, has been addressed.
> >
> > I will therefore increase my rating to 6.

---

> > > ### Author Response · Authors · 2024-11-26
> > >
> > > Thank you for recognizing our efforts and providing constructive feedback.  We're glad to hear that you are now satisfied with the improvements to our evaluation process, which led to an adjustment in your rating of our manuscript.
> > >
> > > We greatly appreciate your valuable time and thorough review, which significantly helped improve our work.

---

### Official Review · Reviewer_sM5k · 2024-11-05

**Soundness:** 2
**Presentation:** 2
**Contribution:** 2
**Rating:** 5
**Confidence:** 5

**Summary:**

The paper introduces Edge Mutation Differentiable Architecture Search (EM-DARTS), an approach designed to address the performance collapse issue in Differentiable Architecture Search (DARTS).  The authors identify the main causes for collapse performance in DARTS: the coupling between network weights and architecture parameters in the continuous relaxation framework. EM-DARTS introduces a mutation mechanism that alters the DARTS supernet edges during network weight updates, allowing parametric operations to better align with the optimal feature map and reducing the impact of architecture parameters on these operations.

**Strengths:**

1. This paper is overall well-written.

2. Extensive experiments demonstrate that EM-DARTS outperforms some existing methods, including variants of DARTS, across different datasets and search spaces.

3. The edge mutation mechanism introduces negligible computational overhead, preserving the efficiency of DARTS.

**Weaknesses:**

1. I wonder why we need this  edge mutation rather than a simple EM algorithm to decouple the network optimaztion and architecture search.

2. The performance of the proposed approach is not outstanding even compared to methods [R1] [R2] that were proposed three years ago.  Besides, the most recent article cited by the authors is published in 2023. And considering that it was September 2024 at the time of submission, I am curious as to why there was no comparison to the most recent methods, especially those published in 2024.

3.  The effectiveness of EM-DARTS heavily relies on the setting of the mutation probability, which may require careful tuning for different search spaces and datasets.

4. While the paper provides theoretical analysis, the validation largely relies on empirical results. More theoretical insights into the long-term behavior and stability of EM-DARTS could strengthen the paper's contributions.

[R1] Chen, X., Wang, R., Cheng, M., Tang, X., & Hsieh, C. J. (2020). Drnas: Dirichlet neural architecture search. arXiv preprint arXiv:2006.10355.

[R2] Wang, Yaoming, et al. "Learning latent architectural distribution in differentiable neural architecture search via variational information maximization." Proceedings of the IEEE/CVF International Conference on Computer Vision. 2021.

**Questions:**

Please see the weaknesses.

---

> ### Author Response · Authors · 2024-11-24
> **Replying  Weakness 1-2**
>
> We appreciate the reviewer's valuable comments. The following are my responses to the questions you raised:
>
> ### Regarding Weakness 1 (Why choose edge mutation over the traditional EM algorithm):
> The edge mutation mechanism fundamentally differs from the traditional Expectation-Maximization (EM) algorithm. The EM algorithm primarily decouples hidden variables and parameters through iterative optimization, making it suitable for statistical models with hidden variables, such as Gaussian Mixture Models. In the EM algorithm, the E-step (Expectation step) estimates hidden variables by maximizing the posterior probability, while the M-step (Maximization step) updates model parameters by maximizing the likelihood function. This method is highly effective for specific types of models but cannot be directly applied in the context of Neural Architecture Search (NAS), particularly within the DARTS framework.
>
> In contrast, the edge mutation mechanism is an innovative approach introduced within the DARTS framework. In DARTS, the supernet encompasses all possible operations, with each edge representing an operation and weighted by architecture parameters. However, this weighted sum approach leads to coupling between the weights of parametric operations and architecture parameters, causing undertraining of parametric operations and negatively impacting model performance. The edge mutation mechanism introduces randomness during the edge weight updates in the supernet, breaking this coupling. Specifically, each edge has a certain probability $ p $ of mutating into a specific parametric operation during each network weight update, rather than continuing to use the weighted sum of all operations. This mutation allows parametric operations to independently receive input data and perform forward and backward propagation, thereby learning more useful feature representations. As a result, parametric operations can better approximate the optimal feature map, increasing their architecture parameters. Additionally, the edge mutation mechanism retains the efficiency of DARTS. DARTS optimizes network weights and architecture parameters through gradient descent, significantly enhancing search efficiency. The randomness introduced by the edge mutation mechanism does not significantly increase computational overhead, thus maintaining efficient search while addressing the coupling issues present in DARTS, ultimately improving the performance of the final architecture.
>
> ### Regarding Weakness 2 (Limited Performance Improvement)
> On the surface, the performance improvement of EM-DARTS within the current DARTS search space seems insignificant; however, such an enhancement remains quite challenging to achieve. In reality, the architectures discovered by existing methods may have already approached the performance limits of the optimal architecture within the DARTS search space, making any further optimization extraordinarily difficult. Notably, on the CIFAR-10 dataset, our model achieved an accuracy of 97.67% with only 4.4M parameters. Models that exceed 98% accuracy typically have over 100M parameters and often employ advanced techniques such as knowledge transfer, data augmentation, and regularization to boost accuracy.
>
> In this context, EM-DARTS effectively addresses the issue of undertraining parametric operations through the introduction of the edge mutation mechanism, without significantly increasing computational overhead. Although the performance improvement may seem minor, it actually represents a significant advancement within the existing framework. Specifically, EM-DARTS not only enhances the generalization and robustness of the model but also provides a new perspective for understanding and addressing the performance collapse issue in DARTS.
>
> ### Regarding Weakness 2 (Lack of Comparison with Latest 2024 Papers)
> We understand the reviewers' concern about the lack of comparison with the latest 2024 papers. After further investigation, we found that no papers specifically addressing DARTS iternatives  were accepted for publication in 2024. Therefore, we expanded our search to include other categories of NAS papers and discovered that there is limited research experimenting within the DARTS search space. Consequently, we identified a single relevant paper published in 2024, SWAP-NAS, and included its methods, along with [R1] DrNAS and [R2] VIM-NAS, in our comparative experiments. The results show that EM-DARTS still performs well across multiple datasets and search spaces, demonstrating performance improvements compared to these latest methods.

---

> ### Author Response · Authors · 2024-11-24
> **Replying Weakness 3-4**
>
> ### Regarding Weakness 3 (Sensitivity to Mutation Probability)
> The effectiveness of EM-DARTS indeed depends on the setting of the mutation probability, which may require careful tuning for different search spaces and datasets. However, in practical applications, if we do not aim for marginal differences in accuracy of 0.1%-0.2%, the setting of the mutation probability can be quite flexible. In fact, our adjustment of $p_{\max} $ is not aimed at finding the best $ p_{\max} $ value among multiple settings based on the performance of the searched architectures, but rather ensuring that its size is just right to suppress performance collapse. Depending on the needs to prevent performance collapse in different spaces and datasets, we have made targeted adjustments to the setting of $p_{\max}$. Through such adjustments, we ensure that the value of $ p_{\max}$ is just right to achieve the effect of suppressing performance collapse, meaning that the number of skip connections in the searched cell architectures will not exceed two.
>
> ### Regarding Weakness 4 (Insufficient Theoretical Analysis):
>
> The long-term behavior and stability of EM-DARTS are issues of critical importance and complexity. In our research, we have not only established a theoretical foundation in Theorem 1, which proves that EM-DARTS can more effectively train parametric operations upon convergence, but we have also further validated its stability and long-term behavior through a series of meticulously designed experiments.
>
> These experiments span across various search spaces and datasets, with in-depth evaluations of model performance under different hyperparameter configurations. The experimental results consistently show that, regardless of changing conditions, EM-DARTS maintains excellent performance. This not only demonstrates its adaptability in diverse environments but also indirectly confirms its long-term stability and practical effectiveness.

---

> ### Author Response · Authors · 2024-12-02
>
> Dear Reviewer,
>
> Firstly, thank you for your detailed response to our manuscript. It is evident that you have spent considerable time carefully reviewing our work, and we deeply appreciate your attention and valuable feedback.
>
> ### Regarding the Current Research Context
>
> Indeed, in recent years, performance improvements in the field of Neural Architecture Search (NAS) have not been as significant as in earlier stages, especially concerning advancements within the DARTS framework. The literature published in 2024 almost entirely lacks in-depth explorations in this direction. This is because achieving new breakthroughs from existing improvement strategies has proven challenging. However, by stepping outside traditional perspectives on improving DARTS, we have identified a key reason for performance degradation: under the continuous relaxation framework of DARTS, parameter operations are inadequately trained, which affects the accuracy with which architecture parameters reflect the importance of each operation. This discovery is the core message we wish to convey to the academic community. The EM-DARTS method is merely an initial attempt based on this finding, and we firmly believe that future researchers can build upon our insights to propose more advanced solutions.
>
> ### The Two-Stage  of DARTS Tasks
>
> To clarify  DARTS tasks more clearly, allow me to elaborate on its two-stage essence:
>
> 1. **Search Stage**: During this phase, the objective is to find the optimal architecture within a given search space. The parameter size of architectures during this process is uncontrollable and is not due to any deliberate choice favoring larger architectures. In fact, our approach does not exhibit any preference for parameter operations (this point was specifically addressed in my response to Reviewer 8UyK).
>
> 2. **Evaluation Stage**: Once the optimal architecture is determined, we construct an evaluation network based on it for testing. To ensure fairness in evaluation, all conditions except for the architecture itself—such as initial channel numbers and layer counts—are kept consistent. Therefore, any performance improvements should be attributed primarily to architectural optimization rather than changes in parameter size. We discussed this point in detail in our response to Reviewer QBxG and provided an example of an architecture with 5.1M parameters but only 96.87% accuracy, demonstrating that parameter size is not the decisive factor for performance.
>
> ### On the Tuning of Hyperparameter Mutation Probability
>
> Regarding the adjustment of the hyperparameter mutation probability, fine-tuning is indeed necessary. This value is directly related to mitigating performance collapse. When the mutation probability is high, although it effectively suppresses performance collapse, it may adversely affect the training of parameter operations. Conversely, if the mutation probability is too low, it might fail to prevent performance collapse. Thus, setting an appropriate mutation probability is crucial for maintaining model performance. We acknowledge this limitation of our method and explicitly pointed it out in the manuscript's conclusion, proposing it as a direction for future research.
>
> ### Our Main Contribution
>
> We believe that the most significant contribution of this study lies in identifying the issue of inadequate training of parameter operations under the continuous relaxation framework, which leads to performance collapse, and providing thorough theoretical proof for this insight. The EM-DARTS method represents a compromise solution aimed at addressing this problem without significantly increasing search costs. We sincerely hope that future researchers can leverage our findings to develop more robust methods, thereby advancing the field.
>
> Thank you once again for your understanding and support.
>
> Best regards,
>
> The Authors of Paper 5748
>
> December 2, 2024

---

### Meta-Review · Area_Chair_DVoK · 2024-12-24

**Metareview:**

This study introduces EM-DARTS, a method aimed at addressing performance collapse in DARTS through an edge mutation mechanism to enhance parametric operation training and decouple network weights from architecture parameters. While the motivation and theoretical analysis are clear, reviewers found the contributions incremental, with strong similarities to existing methods such as DS-NAS and SPOS. Performance improvements were marginal, and the evaluation fairness and experimental comparisons were questioned. Although the authors provided clarifications and additional experiments, the issues of novelty, hyperparameter sensitivity, and limited comparisons were not fully resolved. The AC agrees with the reviewers and recommends rejection.

**Additional Comments On Reviewer Discussion:**

Reviewers raised consistent concerns regarding the incremental nature of the contributions, hyperparameter sensitivity, and insufficient comparisons to recent NAS methods. Although the authors provided further explanations and experiments, these did not sufficiently address the reviewers' concerns.

---

### Decision · Program_Chairs · 2025-01-22

Reject